# The dynamic interplay of PIP$_2$ and ATP in the regulation of the K$_{ATP}$ channel

Tanadet Pipatpolkai[1,2,3,4] ⓘD, Samuel G. Usher[1,3,5] ⓘD, Natascia Vedovato[1] ⓘD, Frances M. Ashcroft[1] ⓘD and Phillip J. Stansfeld[6,7] ⓘD

[1]*Department of Physiology, Anatomy and Genetics, University of Oxford, Oxford, Oxfordshire, UK*
[2]*Department of Biochemistry, University of Oxford, Oxford, Oxfordshire, UK*
[3]*OXION Initiative in Ion Channels and Disease, University of Oxford, Oxford, Oxfordshire, UK*
[4]*Science for Life Laboratory, Department of Applied Physics, KTH Royal Institute of Technology, Solna, Sweden*
[5]*Department of Drug Design and Pharmacology, University of Copenhagen, Copenhagen, Denmark*
[6]*School of Life Sciences, University of Warwick, Coventry, Warwickshire, UK*
[7]*Department of Chemistry, University of Warwick, Coventry, Warwickshire, UK*

Handling Editors: Ian Forsythe & Thomas DeCoursey

The peer review history is available in the Supporting Information section of this article (https://doi.org/10.1113/JP283345#support-information-section).

*The Journal of Physiology*

**Abstract** ATP-sensitive potassium (K$_{ATP}$) channels couple the intracellular ATP concentration to insulin secretion. K$_{ATP}$ channel activity is inhibited by ATP binding to the Kir6.2 tetramer and activated by phosphatidylinositol 4,5-bisphosphate (PIP$_2$). Here, we use molecular dynamics simulation, electrophysiology and fluorescence spectroscopy to show that ATP and PIP$_2$ occupy different binding pockets that share a single amino acid residue, K39. When both ligands are present, simulations suggest that K39 shows a greater preference to co-ordinate with PIP$_2$ than with ATP. They also predict that a neonatal diabetes mutation at K39 (K39R) increases the number of hydrogen bonds formed between K39 and PIP$_2$, potentially accounting for the reduced ATP

S. G. Usher, N. Vedovato, F. M. Ashcroft, and P. J. Stansfeld contributed equally to the work.

This article was first published as a preprint. Pipatpolkai T, Usher SG, Vedovato N, Ashcroft FM, Stansfeld PJ. 2021. The dynamic interplay of PIP2 and ATP in the regulation of the KATP channel. bioRxiv. https://doi.org/10.1101/2021.05.06.442933.

inhibition observed in electrophysiological experiments. Our work suggests that $PIP_2$ and ATP interact allosterically to regulate $K_{ATP}$ channel activity.

(Received 19 May 2022; accepted after revision 4 August 2022; first published online 1 September 2022)

**Corresponding authors** Frances M. Ashcroft: Department of Physiology, Anatomy and Genetics, University of Oxford, South Parks Road, Oxford, OX1 3QX, UK. Email: Frances.ashcroft@dpag.oxac.uk and Phillip J. Stansfeld: Interdisciplinary Biomedical Research Building, School of Life Sciences, University of Warwick, Coventry CV4 7AL, UK. Email: phillip.stansfeld@warwick.ac.uk

**Abstract figure legend** In this study, we have used electrophysiology, patch-clamp fluorometry and molecular dynamics simulations to study the dynamic interplay of phosphatidylinositol 4,5-bisphosphate ($PIP_2$) and ATP in the regulation of the $K_{ATP}$ channel, identifying K39 as a residue that engages with both ligands.

### Key points

- The $K_{ATP}$ channel is activated by the binding of phosphatidylinositol 4,5-bisphosphate ($PIP_2$) lipids and inactivated by the binding of ATP.
- K39 has the potential to bind to both $PIP_2$ and ATP. A mutation to this residue (K39R) results in neonatal diabetes.
- This study uses patch-clamp fluorometry, electrophysiology and molecular dynamics simulation.
- We show that $PIP_2$ competes with ATP for K39, and this reduces channel inhibition by ATP.
- We show that K39R increases channel affinity to $PIP_2$ by increasing the number of hydrogen bonds with $PIP_2$, when compared with the wild-type K39. This therefore decreases $K_{ATP}$ channel inhibition by ATP.

## Introduction

Pancreatic ATP-sensitive potassium ($K_{ATP}$) channels couple the metabolic state of the pancreatic $\beta$-cell to insulin secretion (Rorsman & Ashcroft, 2018). Cryo-electron microscopy (cryo-EM) studies have provided high-resolution structures of the $K_{ATP}$ channel complex. It consists of a central tetrameric pore formed from four inwardly rectifying potassium channel (Kir6.2) subunits, surrounded by four regulatory sulphonylurea receptor 1 (SUR1) subunits (Puljung, 2018). Binding of ATP to Kir6.2 closes the channel (Tucker et al., 1997), whereas binding of phosphoinositides, such as phosphatidylinositol 4,5-bisphosphate ($PIP_2$), increases the channel open probability (Baukrowitz et al., 1998). The ATP binding site on Kir6.2 has been identified in several cryo-EM structures (Ding et al., 2019; Lee et al., 2017a; Li et al., 2017; Martin et al., 2019; Martin, Kandasamy et al., 2017; Martin, Yoshioka et al., 2017; Wu et al., 2018). The $PIP_2$ binding site has been resolved in structural studies of related Kir channels (Kir2.2 and Kir3.2) (Hansen et al., 2011; Whorton & MacKinnon, 2011) but not yet for Kir6.2. However, the structure of the Kir6.2 $PIP_2$ binding site has been predicted previously using site-directed mutagenesis, docking and coarse-grained molecular dynamics (CG-MD) simulation (Haider et al., 2007; Pipatpolkai, Corey et al., 2020; Shyng et al., 2000; Stansfeld et al., 2009).

Mutations in the $K_{ATP}$ channel lead to diseases of insulin secretion. Channel hyperactivation is associated with reduced insulin secretion and neonatal diabetes mellitus, whereas reduced channel activity leads to enhanced insulin secretion and congenital

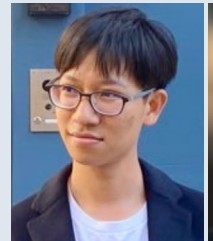
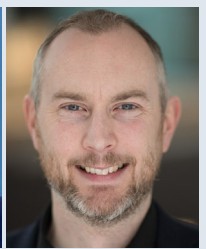

**Tanadet Pipatpolkai** was a DPhil. student, jointly between the Ashcroft and Stansfeld research groups at the University of Oxford, and is now a postdoctoral researcher in the Delemotte laboratory, Stockholm. His research focuses on understanding the role of lipids in ion channel gating and their involvement in disease. Tanadet uses a wide range of molecular dynamics simulation methods to assess the lipid binding sites, free energies and the associated conformational change induced by binding. In his free time, he enjoys baking and cooking a variety of desserts! **Phillip J. Stansfeld** is a Professor in Computational Biochemistry at the University of Warwick, a joint appointment between the School of Life Sciences and the Department of Chemistry. His group applies molecular dynamics simulations and moleculecular modelling techniques to study membrane protein structures, with especial interest in how lipid interactions modulate protein function.

hyperinsulinism (Ashcroft, 2005). Previous studies have shown that many neonatal diabetes mellitus mutations cluster at the ATP binding site, disrupting the ATP sensitivity of the channel (Pipatpolkai, Usher et al., 2020). Some neonatal diabetes mellitus mutations also interfere with PIP$_2$ regulation; for example, E179K enhances PIP$_2$ stimulation of the K$_{ATP}$ current (as indicated by a reduction in neomycin block) and increases the predicted PIP$_2$ binding affinity (De Franco et al., 2020; Pipatpolkai, Corey et al., 2020).

In addition to increasing the open probability of the K$_{ATP}$ channel, PIP$_2$ binding to Kir6.2 reduces the inhibitory effect of ATP on the K$_{ATP}$ current (Fan & Makielski, 1999). The precise mechanisms by which this phenomenon occurs have not been fully resolved. An increase in K$_{ATP}$ channel open probability ($P_{open}$) leads to reduced ATP inhibition (Enkvetchakul et al., 2000; Trapp et al., 1998); therefore, at least part of the effect of PIP$_2$ on ATP inhibition is mediated via changes in $P_{open}$ (Enkvetchakul et al., 2000). However, it has also been argued that PIP$_2$ might have an additional effect on ATP sensitivity that is independent of $P_{open}$ (Enkvetchakul et al., 2000). Both molecules carry similar negatively charged phosphate groups, and previous studies have proposed that PIP$_2$ competes with ATP for binding to the C-terminus of Kir1.1 channels and competes with trinitrophenyl adenosine triphosphate (TNP-ATP) binding to the C-terminus of Kir6.1 and Kir6.2 channels (MacGregor et al., 2002). Comparison of recent structural studies of the channel with bound ATP, and docking and molecular dynamics simulations with PIP$_2$ suggest that ATP and PIP$_2$ have separate binding pockets (Haider et al., 2005, 2007). Nevertheless, even if the two ligands do not share a binding pocket, it is still possible for ATP and PIP$_2$ to 'compete' for the same Kir6.2 subunit, described as 'negative heterotropic cooperativity' (Enkvetchakul et al., 2001; Enkvetchakul & Nichols, 2003).

In this study, we used atomistic molecular dynamic (AT-MD) simulations to determine the dynamics of K39 when both ATP and PIP$_2$ occupy their respective binding sites. These data show that K39 can co-ordinate with both ATP and PIP$_2$, but with stronger preference for PIP$_2$ when both ligands are present. We used a combination of electrophysiology and fluorescence spectroscopy to assess how mutations that affect PIP$_2$ binding modulate ATP inhibition and nucleotide binding. These support the simulation findings and suggest how the mutations give rise to clinical disease.

## Methods

### Coarse-grained system preparation

We used four molecular complexes in our simulations: (i) the human Kir6.2 model from residue 32 to 352 without SUR1 in the propeller conformation (Protein Data Bank (PDB) entry: 6BAA) (Martin, Kandasamy et al., 2017); (ii) the Kir6.2 model in quatrefoil conformation (PDB entry: 6C3O) (Lee et al., 2017b); (iii) an open state Kir6.2 channel with mutation on G334D and C166S (PDB entry: 7S5T); and (iv) a full K$_{ATP}$ propeller channel octameric complex consisting of four copies of Kir6.2 (residues 32−352) and four copies of SUR1 (PDB entry: 6BAA). These complexes were converted to a coarse-grained representation using *martinize.py*, embedded in a palmitoyl-oleoyl-phosphatidylcholine (POPC) bilayer and solvated in water and 0.15 M NaCl using a self-assembly MemProtMD pipeline (Stansfeld et al., 2015; https://github.com/pstansfeld/MemProtMD/). All simulations were carried out using the Martini v.2.2 biomolecular forcefield (Marrink et al., 2007; Monticelli et al., 2008). The tertiary and quaternary structures of the protein were maintained through the application of an elastic network with a force constant of 1000 kJ mol$^{-1}$ nm$^{-2}$ between two coarse-grained backbone particles within 0.5–0.9 nm. A temperature of 323 K was maintained with V-rescale temperature coupling (Bussi et al., 2007), while 1 bar pressure was controlled using semi-isotropic Parrinello–Rahman pressure coupling (Parrinello & Rahman, 1981). The position of the coarse-grained PIP$_2$ is taken from the chicken Kir2.2-PIP$_2$:diC8 after conversion to a coarse-grained model (Hansen et al., 2011). Systems were energy minimized using the steepest descents algorithm and equilibrated for 1 $\mu$s. All simulations were carried out using GROMACS-2019.4 (Van Der Spoel et al., 2005).

### Atomistic simulation set-up

The coarse-grained simulation system (Kir6.2, lipids, PIP$_2$ and POPC, ions and water) was converted to atomistic using the CG2AT pipeline (Stansfeld & Sansom, 2011). The K39R mutant model was generated using PyMOL with the minimum initial stearic clashes (Schrodinger LLC, 2015). The exact position of the ATP molecule was taken from the cryo-EM structure (PDB entry: 6BAA) and placed in the Kir6.2 ATP binding site. The initial position of the hydrogen atoms was added using PyMOL. All simulations were carried out using CHARMM36m biomolecular forcefield with the virtual sites on the CH$_3$ and NH$_3^+$ groups of the proteins and lipids, allowing the integration time step of 4 fs in the production run (Huang et al., 2017; Olesen et al., 2018). The forcefield for the ATP molecule was derived from the CHARMM-GUI (Kim et al., 2017). In this study, four different conditions were set (Apo; ATP bound; PIP$_2$ bound; and both ATP and PIP$_2$ bound), and simulations were carried out for three repeats.

The systems were energy minimized using the steepest descents algorithm, with non-hydrogen atoms restrained

at 1000 kJ mol$^{-1}$ nm$^{-2}$. This was then followed by a 5 ns equilibration for the system where the C$_\alpha$ backbone on the protein and the non-hydrogen atoms on the ATP molecules were restrained with 1000 kJ mol$^{-1}$ nm$^{-2}$ with 4 fs time steps. A temperature of 310 K was maintained with V-rescale temperature coupling (Bussi et al., 2007), while 1 atm pressure was controlled using semi-isotropic Parrinello–Rahman pressure coupling (Parrinello & Rahman, 1981). The simulation was then equilibrated further with only C$_\alpha$ restraint on the protein for another 15 ns in similar conditions. Then three repeats of the 380 ns production runs were implemented, in which the first 80 ns of the simulations was discarded as equilibration. All simulations, hydrogen bond (H-bond) calculations and distance calculation were carried out using GROMACS-2019.4 (Abraham et al., 2015). We used MDAnalysis to calculate pairwise root mean square deviation (RMSD) to define the change in protein dynamics across the trajectory (Beckstein et al., 2009; Gowers et al., 2016; Michaud-Agrawal et al., 2011; Theobald, 2005).

## Molecular biology

Human Kir6.2 and SUR1 were subcloned into pCGFP_EU [green fluorescent proten (GFP)-tagged constructs] and pcDNA4/TO, respectively. Site-directed mutagenesis and amber stop codons were introduced using the QuikChange XL system (Stratagene, San Diego, CA, USA) and verified by sequencing (DNA Sequencing and Services, Dundee, UK), as previously described (Usher et al., 2020). HEK293T cells were grown in in Dulbecco's modified Eagle's medium (DMEM; Sigma) with the addition of 10% fetal bovine serum, 100 U ml$^{-1}$ penicillin and 100 $\mu$g ml$^{-1}$ streptomycin (Thermo Fisher Scientific; Waltham, MA, USA) at 37°C, 5:95 CO$_2$:air. Cells were seeded in T25 flasks for 24 h, before transfection with TransIT-LT1 (Mirus Bio LLC, Madison, WI, USA). Protein expression and trafficking to the plasma membrane were optimized by including 300 $\mu$M tolbutamide in the transfection media (Yan et al., 2007). 3-(6-Acetylnaphthalen-2-ylamino)-2-aminopropanoic acid (ANAP)-tagged Kir6.2 constructs were cultured in the presence of 20 $\mu$M ANAP (free acid; AsisChem, Waltham, MA, USA), and 48 h post-transfection the cells were re-plated onto either poly-D-lysine-coated glass-bottomed FluoroDishes (FD35-PDL-100; World Precision Instruments) or poly-L-lysine-coated 35 mm Petri dishes (Corning). pCDNA4/TO and pANAP were obtained from Addgene. peRF1-E55D (*Homo sapiens*) and pCGFP_EU (*Aequorea victoria*) were kind gifts from the Chin Laboratory (MRC Laboratory of Molecular Biology, Cambridge, UK) and Gouaux Laboratory (Vollum Institute, Portland, OR, USA), respectively.

## Electrophysiology

Extracellular (pipette) solutions contained (mM): 140 KCl, 1 EGTA and 10 Hepes (pH adjusted to 7.3 with KOH). The intracellular (bath) solutions contained (mM): 140 KCl, 1 EDTA, 1 EGTA, 10 Hepes (pH adjusted to 7.3 with KOH). Inside-out patches were excised from transfected HEK293T cells using borosilicate glass pipettes (GC150F-15; Harvard Apparatus, Holliston, MA, USA) pulled to a resistance of 1–3 MΩ. Data were acquired at a holding potential of −60 mV using an Axopatch 200B amplifier and a Digidata 1322A digitizer run through pClamp 9 software (Molecular Devices, San Jose, CA, USA). Currents were low-pass filtered at 1–5 kHz and digitized at 10–20 kHz.

Patches were perfused with an eight-channel $\mu$Flow or a manual gravity perfusion system. Different concentrations of ATP (Sigma) or fluorescent trinitrophenyl adenosine triphosphate (TNP-ATP; Jena Bioscience) were added to the bath solution to assess current inhibition and/or nucleotide binding. Nucleotide-induced inhibition was corrected for rundown by alternating test concentrations of nucleotide solution with nucleotide-free solution. The inhibition was expressed as a fraction of the control currents before and after the test solution, as described previously (Proks et al., 2010). For experiments with TNP-ATP, the zero-current level was determined by perfusing 10 mM BaCl$_2$ at the end of each recording at a holding potential of +60 mV. Current inhibition data were fitted with the following Hill equation:

$$\frac{I}{I_{\max}} = 1 - I_{\max} + \frac{I_{\max}}{1 + 10^{(IC_{50} - [nucleotide]) \times -h}}$$

Fitting was performed with the *brms* (Bayesian Regression Models using 'Stan') package in R as a mixed-effects model (Bürkner, 2017), with the IC$_{50}$ value allowed to vary between individual excised patches. Prior probability distributions were supplied for each parameter as follows:

$$h \sim \text{Normal} \left( \mu : 1, \ \sigma^2 : 0.3 \right)$$

$$I_{\max} \sim \text{Uniform} (\min : 0, \ \max : 1)$$

$$IC_{50} \sim \text{Normal} \left( \mu : -4, \ \sigma^2 : 1 \right)$$

Each model was run across four chains for 4000 iterations including a burn-in period of 2000 iterations for a total of 8000 samples. Each model parameter achieved a minimum effective sample size of 5000 and a potential scale reduction statistic (R̂) of 1.00. Contrasts were calculated by subtracting the full posterior probability for the IC$_{50}$ of the 'control' construct from each compared mutant channel. The resulting probability distribution represents the estimated fold-change in IC$_{50}$ (given that the estimated IC$_{50}$ is expressed as a logarithmic value) as a result of making the indicated mutation.

## Fluorescence measurements

Fluorescence spectra from excised patches were collected and analysed as described previously (Usher et al., 2020). Briefly, the tip of the patch pipette was centred on the slit of the spectrometer immediately after patch excision. ANAP was excited using a 385 nm LED source (ThorLabs, Newton, NJ, USA) with a 390 nm/18 nm bandpass excitation filter, after which the emitted light passed through a 400 nm long-pass emission filter (ThorLabs) and an IsoPlane 160 Spectrometer (Princeton Instruments, Trenton, NJ, USA) with a 300 groove mm$^{-1}$ grating. Images were collected with 1 s exposures on a Pixis 400BR_eXcelon CCD (Princeton Instruments). Spectra were corrected for background fluorescence, then ANAP intensity was calculated by averaging the fluorescence intensity measured between 469.5 and 474.5 nm. This intensity was corrected for bleaching with a single exponential decay.

## Statistics and data presentation

Concentration–response data are plotted, with each measurement shown as a data point. The Hill equation fits for nucleotide inhibition are displayed as the median fit (continuous line) and the 95% intervals (shaded area) of the posterior probability distribution. The Monod–Wyman–Changeux (MWC)-type model fits are also displayed as the median fit (continuous line) and the 95% intervals (shaded area) of the posterior probability distribution. Where posterior probability distributions for parameter values are shown, they are displayed with the 50, 80 and 95% intervals of the distribution in progressively lighter shades of colour.

## Monod–Wyman–Changeux model fitting

The MWC-type model fitted here is described by the following sets of equations:

to open and close in the absence of nucleotides. Each ligand binding event ($K_A$) is independent, and each bound ligand favours the closed state by the same factor ($D$). This model is fitted to the combined current inhibition and fluorescence quenching data using the *brms* package in R (Bürkner, 2017). Prior probability distributions were supplied for each parameter as follows:

$$\log_{10}(L) \sim \text{Normal}\left(\mu : 0, \ \sigma^2 : 0.7\right)$$

$$\log_{10}(K_A) \sim \text{Uniform}\left(\min : 2, \ \max : 6\right)$$

$$D \sim \text{Uniform}\left(\min : 0, \ \max : 1\right)$$

The model was run across four chains for 4000 iterations including a burn-in period of 2000 iterations for a total of 8000 samples. Each model parameter achieved a minimum effective sample size of 5000 and a potential scale reduction statistic ($\hat{R}$) of 1.00.

## Results

In this study, we used both atomistic molecular dynamic simulations and functional studies to explore the relationship between the ATP and PIP$_2$ binding sites of Kir6.2.

### Simulation studies: exploring the PIP$_2$ and ATP binding sites on Kir6.2

We simulated the PIP$_2$ and ATP binding sites in the Kir6.2 tetramer in the absence of SUR1 in order to exclude ATP interactions with SUR1. Previous simulations have shown that there is no difference in the PIP$_2$ binding site when SUR1 is present (Pipatpolkai, Corey et al., 2020). In support of this observation, PIP$_2$ induces an increase in the $P_{\text{open}}$ of Kir6.2 even in the absence of SUR1 (Enkvetchakul et al., 2000; Fan & Makielski, 1999;). Although the structure of the wild-type (WT) Kir6.2/SUR1 octameric complex has been resolved (Ding et al., 2019; Lee et al., 2017a; Li et al., 2017; Martin,

$$\frac{F}{F_{\max}} = \frac{K_A \, [\text{TNP-ATP}] \times (1 + K_A \, [\text{TNP-ATP}])^3 + LDK_A \, [\text{TNP-ATP}] \times (1 + DK_A \, [\text{TNP-ATP}])^3}{(1 + K_A \, [\text{TNP-ATP}])^4 + L(1 + DK_A \, [\text{TNP-ATP}])^4}$$

$$\frac{I}{I_{\max}} = \frac{L(1 + DK_A \, [\text{TNP-ATP}])^4}{(1 + K_A \, [\text{TNP-ATP}])^4 + L(1 + DK_A \, [\text{TNP-ATP}])^4} \times \frac{1 + L}{L}$$

The full rationale behind this model choice was described in more detail previously (Puljung et al., 2019; Usher et al., 2020). Briefly, in this model, $L$ represents an equilibrium constant where the K$_{ATP}$ open probability ($P_{\text{open}}$) is equal to $L/(L + 1)$, reflecting the ability of K$_{ATP}$

Kandasamy et al., 2017; Martin, Yoshioka et al., 2017; Martin et al., 2019; Wu et al., 2018), there is no structure of Kir6.2 in either a PIP$_2$-bound or an ATP-bound conformation when SUR1 is absent. Thus, we built two atomistic simulation systems: Kir6.2 with ATP and Kir6.2

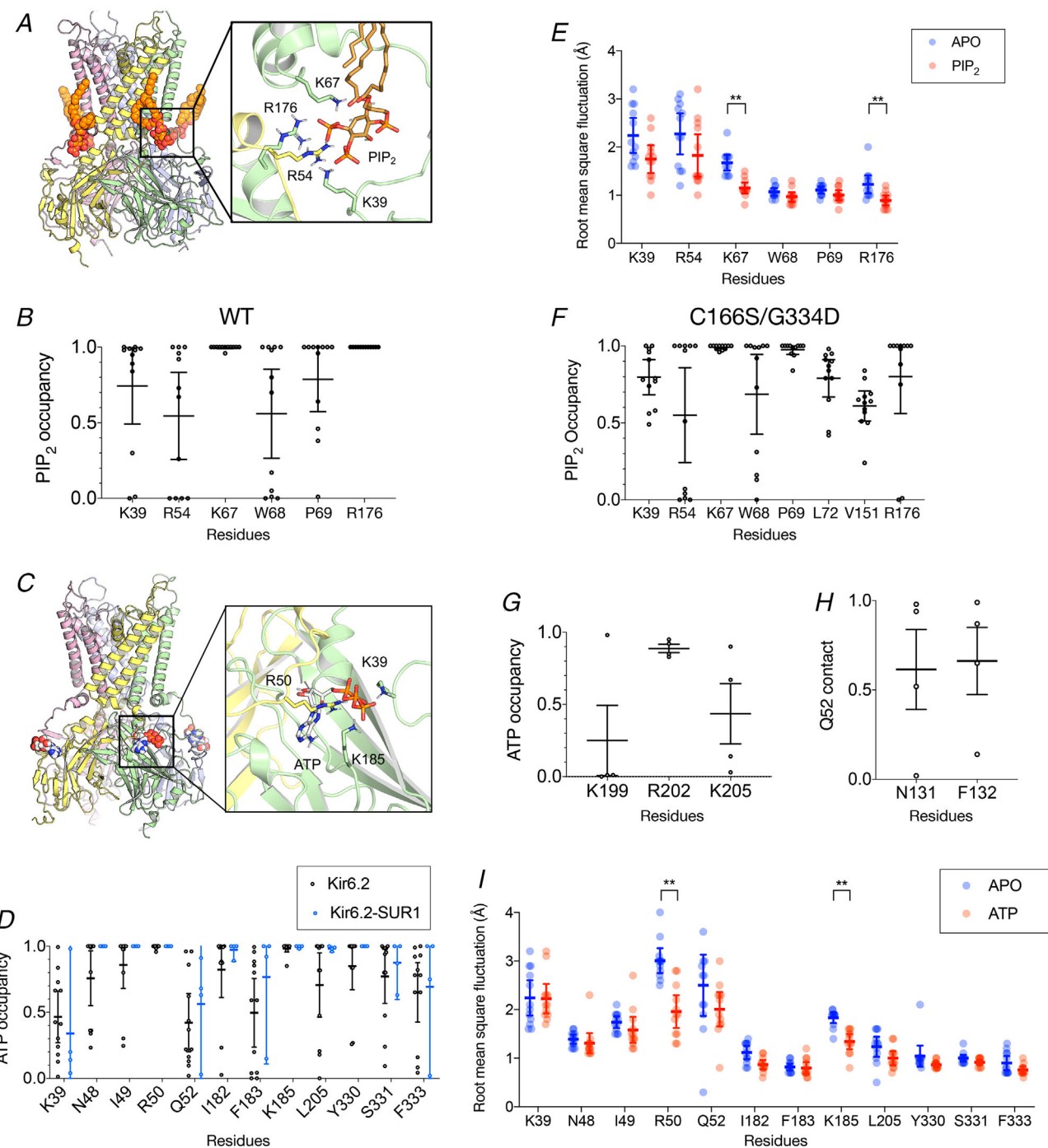

**Figure 1. Phosphatidylinositol 4,5-bisphosphate binding sites**

*A*, the structure of the Kir6.2 tetramer shown with four phosphatidylinositol 4,5-bisphosphate (PIP₂) molecules in their binding sites. The inset shows interactions between PIP₂ (bronze) and basic residues in two chains of Kir6.2 (yellow and green). *B*, the fraction of time that the indicated residues are in <4 Å proximity to PIP₂ during the final 300 ns of the simulations (defined as 'PIP₂ occupancy'). *C*, the structure of the Kir6.2 tetramer shown with four ATP molecules in their binding site. The inset shows interactions between ATP (Corey-Pauling-Koltun (CPK) space-filling representation) and basic residues in two chains of Kir6.2 (yellow and green) after a 380 ns simulation. *D*, the fraction of time that the indicated residues are in <4 Å proximity to ATP during the final 300 ns of the simulations (defined as 'ATP occupancy'). *E*, root mean square fluctuation analysis of the residues on the Kir6.2 tetramer that contact the PIP₂ molecule in the absence (blue) and the presence (red) of PIP₂. *F*, the PIP₂ occupancy during the final 300 ns of the simulations of the C166S/G334D structure that was captured in an 'open state'. *G*, the fraction of time that the indicated residues on SUR1 that are in <4 Å proximity to ATP

(defined as 'ATP occupancy') collected during 100 ns of the simulation for Kir6.2+SUR1 simulation (blue). *H*, the fraction of time that the indicated residues on SUR1 that are in <4 Å proximity to Q52 on Kir6.2 (defined as 'Q52 contact') collected during 100 ns of the simulation for Kir6.2+SUR1 simulation (blue). *I*, root mean square fluctuation analysis of the residues on the Kir6.2 tetramer that contact the ATP molecule in the absence (blue) and the presence (red) of ATP. For each plot, three simulations were run, and each subunit of the tetramer is treated as an individual data point (for a total of 12 data points). For each occupancy data point, only residues with >0.4 occupancy are shown. The error bar indicates the 95% confidence interval around the mean. **$P < 0.01$ (Student's unpaired *t*-test). [Colour figure can be viewed at wileyonlinelibrary.com]

with PIP$_2$, and simulated each for 380 ns. To ensure that the initial protein structure was stable after converting to an atomistic system, we used pairwise RMSD analysis over the last 300 ns of the simulation on the C$_\alpha$ atom of Kir6.2 as a measure of the stability of the tertiary structure of the protein. We observed that the C$_\alpha$ RMSD never deviated by >4 Å across the trajectory in all simulation set-ups. Interestingly, the simulations with bound ligand (PIP$_2$ or ATP) showed slightly less C$_\alpha$ rearrangement than the Apo state. This suggests that the three-dimensional structure of the protein is highly stable throughout our simulation and might be further stabilized by the ligand (Fig. 1*A* and *B*). To investigate the effect of PIP$_2$ and ATP binding on the local geometry of their binding sites, we selected amino acid residues within 4 Å of the ligand for >40% of the time and defined them as contacting residues. Between the different Kir6.2 subunits, we found no significant difference in the contact that these residues made with either ATP or PIP$_2$ in the final 300 ns of the simulation. Thus, in all subsequent analyses, we defined each Kir6.2 subunit as a separate calculation to increase the sampling of an ATP molecule in the binding pocket. Given that there are four subunits and three separate simulations were run, this yielded a total of 12 data points per contacting residue. We found that in the last 300 ns of the simulation, the contacts made with the ligands were in good agreement with previous electrophysiological and computational studies (Fig. 1*A*–*D*; Haider et al., 2007; Shyng et al., 2000; Stansfeld et al., 2009).

In the PIP$_2$ binding site, we found that R54 and K67 from one subunit and R176 from an adjacent subunit co-ordinate with the 4′ phosphate of PIP$_2$, and that K39 co-ordinates with the 5′ phosphate (Fig. 1*A* and *B*). Other uncharged residues (W68 and P69) that lie at the membrane–water interface also make strong contact with PIP$_2$. With the exception of P69, mutations at these residues have previously been shown to alter channel ATP sensitivity, increase the open probability and/or alter PIP$_2$ activation (Cukras et al., 2002; Haider et al., 2007; Männikkö et al., 2011; Shyng et al., 2000). By evaluating the root mean square fluctuation of all contact residues, we found that the binding of PIP$_2$ statistically reduces the dynamics of the K67 and R176 side-chains (Fig. 1*E*). However, only the difference at K67 showed biological significance (defined as a decrease of ∼1 Å or more). These results agree well with previous coarse-grained

simulations and therefore validate the co-ordination geometry of PIP$_2$ in its binding site (Pipatpolkai, Corey et al., 2020; Stansfeld et al., 2009). Recently, the structure of an open-state Kir6.2 channel with G334D and C166S mutations was solved (Zhao & MacKinnon, 2021). We conducted additional simulations to analyse the PIP$_2$ co-ordination geometry in the open-state channel. These simulations suggest that there is no significant difference in the contact profile between the open and closed states of the channel (Fig. 1*F*).

In the ATP binding site, we found that R50 co-ordinates with both the $\beta$ and the $\gamma$ phosphate, K39 co-ordinates with the $\gamma$ phosphate and K185 co-ordinates with the $\alpha$ and $\beta$ phosphate (Fig. 1*C* and *D*). Both R50 and K185 dynamics are stabilized when ATP binds to the channel (Fig. 1*I*). These findings agree with previous studies in which the ATP binding residues in Kir6.2 were mapped using site-directed mutagenesis (Dabrowski et al., 2004; Tucker et al., 1998). Interestingly, the side-chain of K39, which is ∼7 Å from the ATP molecule in the cryo-EM structure, moves towards ATP and makes a contact in some simulations. All residues that contact ATP in the cryo-EM structure of the octameric Kir6.2/SUR1 complex (N48, I49, Q52, I182, F183, L205, Y330, S331 and F333) also make contact in our simulations. These residues have been found to be crucial for ATP inhibition in functional studies (Tucker et al., 1998). Mutations in residues that make contact with ATP also cause neonatal diabetes (Pipatpolkai, Usher et al., 2020).

Previous studies have highlighted the significance of SUR1 in ATP binding in Kir6.2 (Tucker et al., 1997). We therefore conducted a short set of 100 ns simulations of the full K$_{ATP}$ octameric complex. Here, we showed that the contacting residues between ATP and Kir6.2 remain unchanged in the presence of SUR1 (Fig. 1*D*). However, we observed additional contacts between R202 (occupancy = 0.89) and K205 (occupancy = 0.44) in Kir6.2 with SUR1, in agreement with previous electrophysiological studies (Usher et al., 2020) (Fig. 1*G*). We also observed contacts between an ATP-contacting residue, Q52, and SUR1 residues N131 and F132 (Fig. 1*H*). Mutations at F132 on SUR1 to L and V are associated with developmental delay, epilepsy and neonatal diabetes (Ellard et al., 2007; Rafiq et al., 2008). This suggested potential residues where Kir6.2 might couple to SUR1 during the gating transitions.

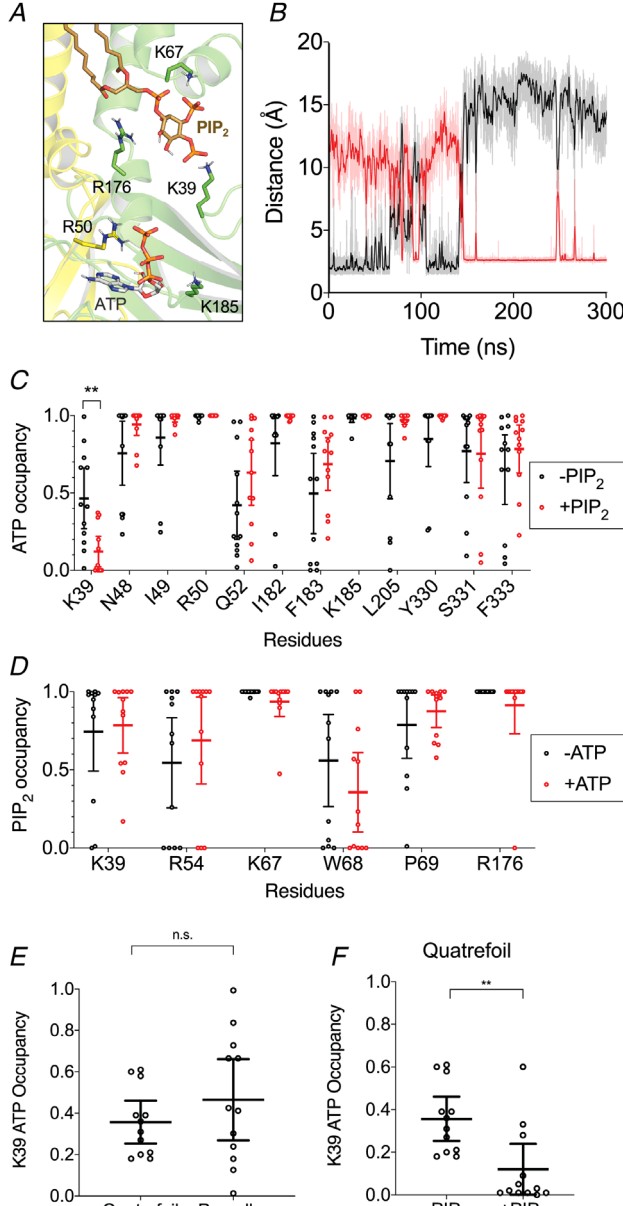

structures. Only residues where the mean occupancy is >0.4 are plotted. Three simulations were run, and each subunit of the tetramer is treated as an individual data point (for a total of 12 data points). The error bar indicates the 95% confidence interval around the mean. **$P < 0.01$ (Student's unpaired *t*-test). [Colour figure can be viewed at wileyonlinelibrary.com]

## The competition between PIP₂ and ATP for K39 co-ordination

Previous studies have shown that PIP₂ reduces channel ATP inhibition (Baukrowitz et al., 1998; Fan & Makielski, 1999; Hilgemann & Ball, 1996; Shyng & Nichols, 1998). However, it was not clear whether PIP₂ competes with ATP for the binding to Kir6.2 subunits or whether it interferes with ATP-dependent gating (or both). Comparison of the cryo-EM structure of the ATP binding site with the predicted PIP₂ binding site suggests that they lie ~25 Å from one another (Martin, Kandasamy et al., 2017; Pipatpolkai, Corey et al., 2020). In our studies, K39 contacts both ATP and PIP₂ in independent simulations, but the position of the side-chain amine group is different.

We next explored the dynamics of K39 when both ligands occupied their respective binding sites (Fig. 2*A*). We calculated the distance between ATP or PIP₂ and the side-chain amine of K39. We observed that the position of K39 oscillated between co-ordination with ATP and PIP₂, but favoured PIP₂ more strongly (Fig. 2*A* and *B*). Interestingly, a significant decrease ($P < 0.01$) in ATP contacts occurred at residue K39 only when PIP₂ was present (Fig. 2*C*). Therefore, we hypothesize that K39 might change its co-ordination from ATP to PIP₂ when both molecules are bound. The contact between K39 and PIP₂ remained unchanged in both in the presence and the absence of ATP (Fig. 2*D*). This phenomenon was also observed using the quatrefoil structure of Kir6.2 (PDB ID: 6C3O; Fig. 2*E* and *F*). Thus, we propose that the salt bridges between K39 and ATP are broken in the presence of the PIP₂, causing the side-chain amine group on K39 to swing towards the PIP₂ headgroup.

With the exception of K39, no other residues in either the ATP or PIP₂ binding site altered their contact probabilities when both ligands were present simultaneously in their respective binding sites.

## Computational and electrophysiological assessment of a neonatal diabetes mutation (K39R)

A mutation at K39 (K39R) is associated with transient neonatal diabetes (Zhang et al., 2015). This substitution does not alter the charge of the side-chain (because both lysine and arginine are positively charged), but an amine is replaced with a guanidium group. Given the results of our simulations, we would predict that this would result in an increased affinity for both PIP₂ and ATP.

**Figure 2. Changes in the ATP and phosphatidylinositol 4,5-bisphosphate binding sites when both ligands are present**
*A*, interactions between Kir6.2 (two chains are indicated in yellow and green), ATP (grey CPK colouring) and the phosphatidylinositol 4,5-bisphosphate (PIP₂; bronze) headgroup. Only the basic residues of the protein are shown. *B*, a representative trace showing the calculation of distance between K39 and ATP (black) and between K39 and PIP₂ (red) across a single 300 ns trajectory. The darker lines show running averages of every 1 ns of the simulation. *C*, the fraction of time that the residues are in <4 Å proximity to ATP during the final 300 ns of the simulations (defined as 'ATP occupancy') in the absence (black) and presence (red) of PIP₂. *D*, the fraction of time that the residues are in <4 Å proximity to PIP₂ during the final 300 ns of the simulations (defined as 'PIP₂ occupancy') in the absence (black) and presence (red) of ATP. *E*, the fraction of time that K39 residues on Kir6.2 are in <4 Å proximity to ATP (defined as 'ATP occupancy') in the presence and absence of PIP₂. *F*, the ATP occupancy in the quatrefoil and propeller Kir6.2

We simulated Kir6.2 containing the K39R substitution and compared the contacts between the guanidium group of the arginine and PIP$_2$ or ATP. In simulations with ATP alone, K39R spent more time in contact with ATP than the WT K39 (Fig. 3*A–C*). In simulations with PIP$_2$ alone, residue 39 in both WT and K39R channels spent almost all of its time co-ordinating PIP$_2$. When both ATP and PIP$_2$ were present, we found that the contact probability of residue 39 with the PIP$_2$ headgroup was not significantly different in either the absence or the presence of the ATP (Fig. 3*D* and *E*). We also observed that the presence of PIP$_2$ reduced the contacts between K39R and ATP, similar to the WT channel (Fig. 3*F* and *G*). From this, we conclude that K39R mutation does not affect channel preference for PIP$_2$ in either the presence or the absence of ATP.

To quantify the strength of the interaction between PIP$_2$ and K39 or K39R, we carried out an H-bond analysis to determine the number of H-bonds formed between the PIP$_2$ headgroup and the side-chain of residue 39. Note that arginine can form up to five H-bonds spread over three amine groups, whereas lysine is able to form a maximum of only three bonds from a single amine. We observed that the guanidium group on the arginine formed two H-bonds with the PIP$_2$ headgroup, whereas the lysine amine group formed only a single H-bond. In both cases, the H-bonds were formed with the 5′ phosphate on the PIP$_2$ inositol headgroup in both the presence and the absence of ATP (Fig. 3*D* and *E*). This suggests that the K39R mutation enhances the strength of the interaction of residue 39 with PIP$_2$.

We next calculated changes in an interaction between K39R and ATP. Here, we showed that K39R mutant was significantly more likely to form two H-bonds with ATP (Fig. 3*F*). However, this phenomenon was abolished when PIP$_2$ was present (Fig. 3*G*). Thus, we could propose that the binding of PIP$_2$ overrides the effect of channel inhibition by ATP in both the WT and the K39R mutant. We postulate that this leads to reduced channel inhibition by ATP and that it is the mechanism by which this mutation impairs insulin secretion and thereby leads to neonatal diabetes.

To explore this hypothesis functionally, we introduced three different substitutions at residue K39 (K39A, K39E and K39R) into Kir6.2 and measured ATP inhibition of K$_{ATP}$ currents in excised patches from HEK293T cells (Fig. 4*A*). To facilitate identification of cells expressing the construct of interest, we attached GFP to the C-terminus of each Kir6.2 construct. We also co-transfected cells with SUR1 to allow assembly of fully octameric K$_{ATP}$ channels. Kir6.2 with a C-terminal GFP tag and co-expressed with SUR1 (hereafter referred to as Kir6.2+SUR1) had an estimated IC$_{50}$ value for ATP inhibition of 23.5–43.7 $\mu$M (95% intervals of the posterior probability distribution). We observed that substitution of K39 by either arginine (Kir6.2-K39R+SUR1) or glutate (Kir6.2-K39E+SUR1)

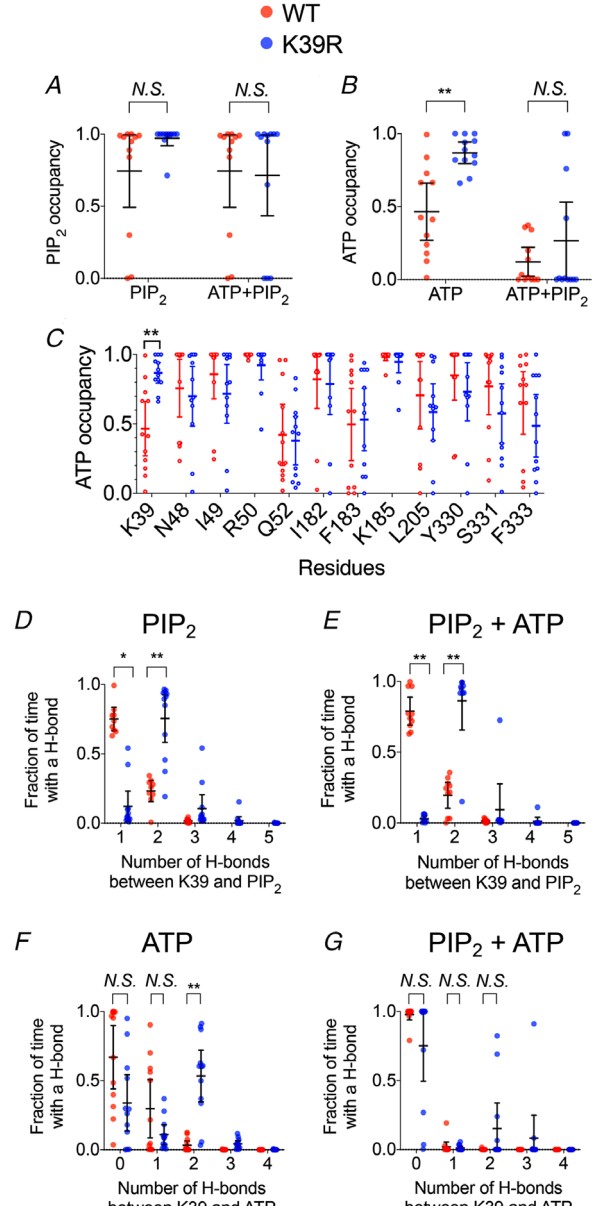

**Figure 3. The K39R mutation changes the phosphatidylinositol 4,5-bisphosphate binding configuration**
The fraction of time that the K39 or K39R are in <4 Å proximity to either ATP or phosphatidylinositol 4,5-bisphosphate (PIP$_2$) during the final 300 ns of the simulations. *A*, the PIP$_2$ occupancy in the absence and presence of ATP. *B*, the ATP occupancy in the absence and presence of PIP$_2$. *C*, the fraction of time that K39 (or K39R) residues are in < 4Å proximity to ATP during the final 300 ns of the simulations (defined as 'ATP occupancy') when both ATP and PIP$_2$ are present. *D–G*, hydrogen bond (H-bond) analysis showing the fraction of time when K39 (red) or K39R (blue) forms a different number of H-bonds with the PIP$_2$ headgroup during the final 300 ns of the simulations in the absence (*D*) and presence of ATP (*E*). Also shown is the number of H-bonds with ATP during the final 300 ns of the simulations in the absence (*F*) and presence of PIP$_2$ (*G*). In all analyses, there are three simulations, and each subunit of a tetramer is treated as an individual data point (a total of *n* = 12). The error bar indicates the 95% confidence interval around the mean. **P < 0.01 (Student's unpaired *t*-test). [Colour figure can be viewed at wileyonlinelibrary.com]

resulted in an increase in the estimated IC$_{50}$ value for ATP inhibition, whereas substitution by alanine (Kir6.2-K39A+SUR1) had no effect (Fig. 4*B* and *C*).

## The effect of substitutions at K39 on nucleotide binding assayed by patch-clamp fluorometry (PCF)

We next employed a recently described fluorimetric technique to directly measure nucleotide binding to the inhibitory binding site on Kir6.2 in the context of different amino acid substitutions at K39 (Puljung et al., 2019; Usher et al., 2020). Briefly, we introduced the fluorescent unnatural amino acid ANAP at residue W311 of Kir6.2 with a C-terminal GFP tag (Kir6.2*). W311 was chosen because it is far away from the ATP binding site and is unlikely to interfere with channel inhibition by TNP-ATP. We then measured Förster resonance energy transfer

(FRET) between ANAP and a fluorescent analogue of ATP, trinitrophenyl-ATP (TNP-ATP), while simultaneously measuring inhibition of K$_{ATP}$ currents (Fig. 5*A*). In the absence of TNP-ATP, excised patches from HEK293T cells expressing Kir6.2*+SUR1 exhibited a fluorescence spectrum with two peaks: one at 470 nm, corresponding to incorporated ANAP, and one at 510 nm, corresponding to GFP. Application of TNP-ATP to excised patches resulted in a concentration-dependent inhibition of K$_{ATP}$ currents and concomitant quenching of the ANAP fluorescence peak in each construct tested.

TNP-ATP inhibited K$_{ATP}$ currents at lower concentrations than ATP for both Kir6.2+SUR1 and Kir6.2*-SUR1; the shift in nucleotide sensitivity was also similar for each of the mutants (A, E and R; Fig. 6). Given that use of this fluorescent analogue is necessary to measure binding to the Kir6.2 nucleotide binding

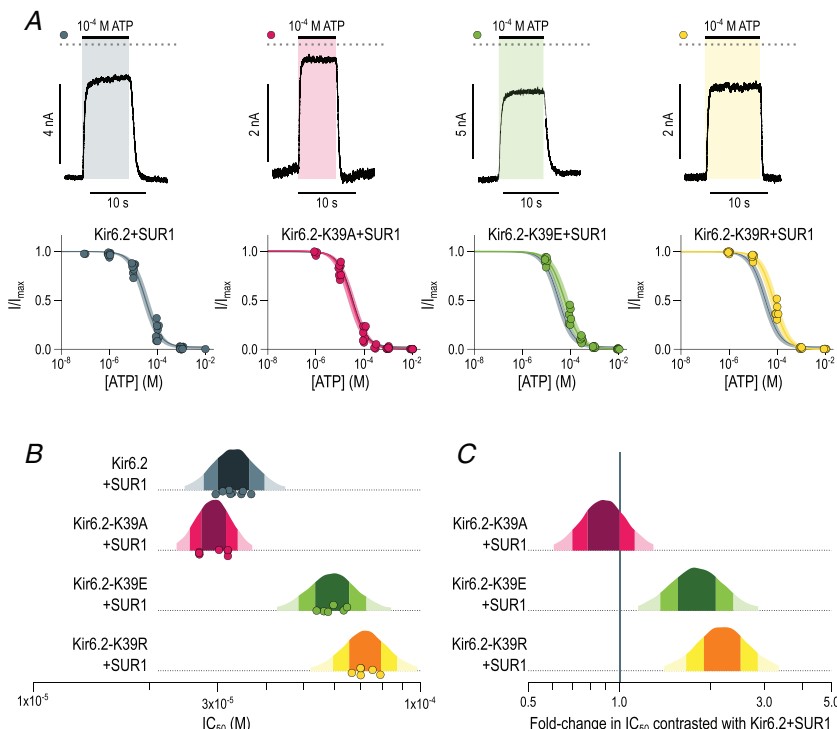

**Figure 4. Substitutions at K39 alter ATP inhibition of K$_{ATP}$ channels**
*A*, ATP concentration–response curves recorded from inside-out patches excised from HEK293T cells co-expressing the indicated green fluorescent protein-tagged Kir6.2 construct and SUR1. Each data point is a single measurement, normalized to the maximum current observed in that patch. The fits are to the Hill equation specified in the Methods, with the continuous line representing the median fit and the shaded area the 95% interval of the posterior probability distribution. The fit for Kir6.2+SUR1 is shown in grey in each panel to facilitate comparison. Parameters for each fit are given in Table 1. A representative current trace showing a typical response to application of 100 $\mu$M ATP is shown above each concentration–response plot. The zero-current level is shown as a dotted line. *B*, posterior probability distributions for the IC$_{50}$ parameters estimated from the Hill fits shown in *A*. The progressively lighter shades of colour represent the 50, 80 and 95% intervals of the distribution. There is a 95% probability that the true IC$_{50}$ value for each construct is located within the lightest coloured interval. The IC$_{50}$ estimates for each patch are shown as filled circles. *C*, fold-change between the indicated IC$_{50}$ value for ATP inhibition of control Kir6.2+SUR1 currents (indicated by the vertical line at 1.0) and that of the indicated mutant channel. Same colour code as in *B*. [Colour figure can be viewed at wileyonlinelibrary.com]

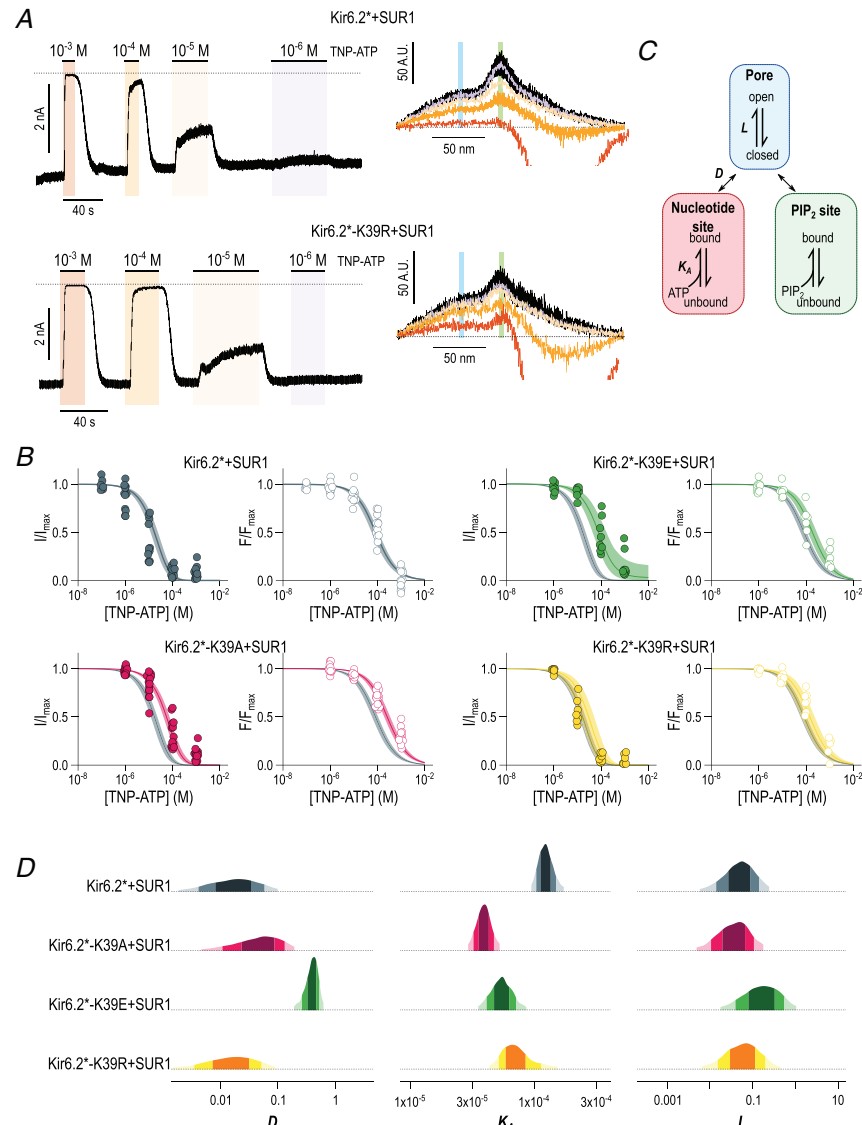

**Figure 5. Substitutions at K39 alter nucleotide binding to Kir6.2**

*A*, representative current (left) and fluorescence (right) traces for Kir6.2*+SUR1 (top) and Kir6.2*-K39R+SUR1 (bottom). The coloured regions of the current traces indicate application of trinitrophenyl adenosine triphosphate (TNP-ATP), with concentrations given as the molarity. The correspondingly coloured spectral traces were captured simultaneously with the current recordings. The light blue region indicated in the fluorescence traces indicates the fluorescence peak corresponding to 3-(6-acetylnaphthalen-2-ylamino)-2-aminopropanoic acid (ANAP), and the light green region that corresponding to green fluorescent protein. The dotted lines indicate the zero-current levels, and the background fluorescence levels. *B*, concentration–response curves for current inhibition (filled data points) and fluorescence quenching (open data points) for the indicated channels. Each data point is a single measurement, normalized to the maximum current or fluorescence observed in that patch. Fluorescence measurements have been corrected further to account for fluorescence bleaching and crosstalk as described in the Methods. The fits are to the Monod–Wyman–Changeux (MWC) equation specified in the Methods, with the continuous line representing the median fit and the shaded area the 95% interval of the posterior probability distribution. The fits for Kir6.2*+SUR1 are shown in grey in each panel to facilitate comparison. *C*, schematic diagram of the MWC-type model used to model the regulation of the K_ATP channel by nucleotide binding to Kir6.2. The three equilibrium parameters used to fit our observed data (*L*, *K*_A and *D*) are shown in bold. Their definitions are given in the main text. *D*, posterior probability distributions for each of the three parameters estimated in the MWC fits shown in *B*. The progressively lighter shades of colour represent the 50, 80 and 95% intervals of the distribution. The probability that the true parameter value of each construct is located within the lightest coloured interval is 95%. [Colour figure can be viewed at wileyonlinelibrary.com]

**Table 1. Inhibition of Kir6.2 constructs by ATP**

| Construct | 95% lower quantile ($\mu$M) | 95% upper quantile ($\mu$M) | Median ($\mu$M) | $n$ |
|---|---|---|---|---|
| Kir6.2 | 94.1 | 455 | 215 | 5 |
| Kir6.2+SUR1 | 24.5 | 43.7 | 32.7 | 8 |
| Kir6.2-K39A+SUR1 | 23.7 | 36.2 | 29.2 | 7 |
| Kir6.2-K39E+SUR1 | 42.7 | 83.0 | 59.2 | 6 |
| Kir6.2-K39R | 159 | 762 | 370 | 3 |
| Kir6.2-K39R+SUR1 | 53.6 | 97.8 | 72.1 | 6 |

Fitted parameters for the Hill fits to dose–response curves shown in Fig. 4*A*. Fitting was performed as described in the Methods. Each *n* is a single inside-out patch from which a complete dose–response has been measured.

site directly, our findings therefore come with the caveat that measurement of TNP-ATP binding might not reflect ATP binding to Kir6.2 directly. Given that $K_{ATP}$ channels are inhibited by a range of physiologically present nucleotides binding to Kir6.2 with different inhibitory strengths (Dabrowski et al., 2003; Trapp et al., 1997), we consider that our findings should be generalizable to other nucleotides.

Given that ANAP is incorporated in a site-specific manner close to the Kir6.2 nucleotide binding site, the

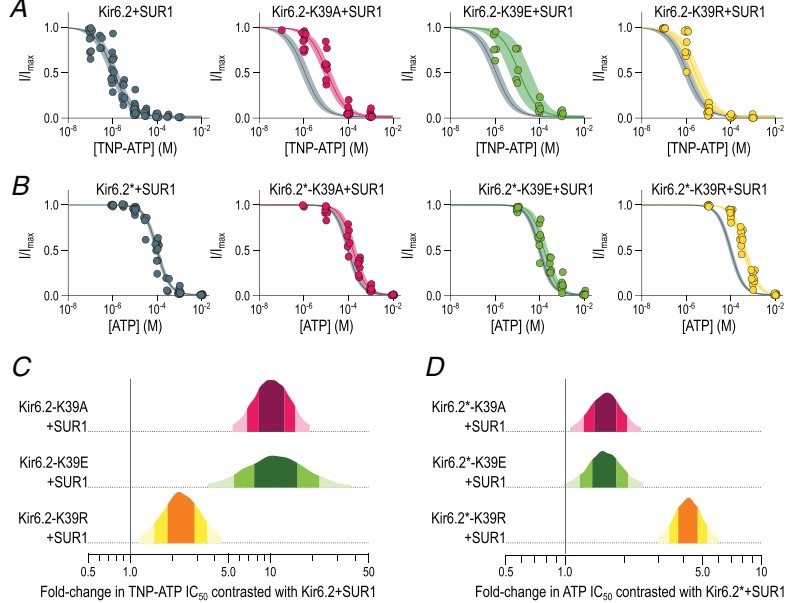

**Figure 6. Substitutions at K39 alter inhibition of $K_{ATP}$ channels by trinitrophenyl adenosine triphosphate**
*A*, trinitrophenyl adenosine triphosphate (TNP-ATP) concentration–response curves recorded from inside-out patches excised from HEK293T cells co-expressing the indicated green fluorescent protein (GFP)-tagged Kir6.2 construct and SUR1. Each data point is a single measurement, normalized to the maximum current observed in that patch. The fits are to the Hill equation specified in the Methods, with the continuous line representing the median fit and the shaded area the 95% interval of the posterior probability distribution. The fit for Kir6.2+SUR1 is shown in grey in each panel to facilitate comparison. Parameters for each fit are given in Table 2. *B*, ATP concentration–response curves recorded from inside-out patches excised from HEK293T cells co-expressing the indicated GFP-tagged Kir6.2* construct and SUR1. Data are presented and analysed as in *A*, and parameters for each fit are given in Table 3. *C*, posterior probability distributions of the fold-change between the $IC_{50}$ value for TNP-ATP inhibition of control Kir6.2+SUR1 currents (indicated by the vertical line at 1.0) and that of the indicated mutant channel. The progressively lighter shades of colour represent the 50, 80 and 95% intervals of the distribution. The probability that the true fold-change in $IC_{50}$ for each construct is located within the lightest coloured interval is 95%. *D*, posterior probability distributions of the fold-change between the $IC_{50}$ value for ATP inhibition of control Kir6.2*+SUR1 currents and that of the indicated mutant channel. The colour scheme is the same as in *C*. [Colour figure can be viewed at wileyonlinelibrary.com]

**Table 2. Inhibition of Kir6.2 constructs by TNP-ATP**

| Construct | 95% lower quantile ($\mu$M) | 95% upper quantile ($\mu$M) | Median ($\mu$M) | *n* |
|---|---|---|---|---|
| Kir6.2+SUR1 | 0.70 | 1.74 | 1.08 | 13 |
| Kir6.2-K39A+SUR1 | 7.06 | 17.0 | 11.0 | 9 |
| Kir6.2-K39E+SUR1 | 4.20 | 39.7 | 11.8 | 5 |
| Kir6.2-K39R+SUR1 | 1.50 | 4.17 | 2.48 | 9 |

Fitted parameters for the Hill fits to dose–response curves shown in Fig. 6*A* and *C*. Fitting was performed as described in the Methods. Each *n* is a single inside-out patch from which a complete dose–response has been measured.

**Table 3. Inhibition of Kir6.2* constructs by ATP**

| Construct | 95% lower quantile ($\mu$M) | 95% upper quantile ($\mu$M) | Median ($\mu$M) | *n* |
|---|---|---|---|---|
| Kir6.2*+SUR1 | 79.6 | 97.7 | 119 | 11 |
| Kir6.2*-K39A+SUR1 | 109 | 157 | 226 | 7 |
| Kir6.2*-K39E+SUR1 | 101 | 154 | 234 | 7 |
| Kir6.2*-K39R+SUR1 | 308 | 412 | 547 | 8 |

Fitted parameters for the Hill fits to dose–response curves shown in Fig. 6*B* and *D*. Fitting was performed as described in the Methods. Each *n* is a single inside-out patch from which a complete dose–response has been measured.

observed fluorescence quenching is directly proportional to the extent of TNP-ATP binding to Kir6.2 (Usher et al., 2020). We therefore fitted the combined current inhibition and fluorescence quenching data to a simple three-parameter MWC-type model (Fig. 5*B*) to identify whether the changes in nucleotide inhibition we observed as a result of mutating K39 can be attributed to a particular biophysical parameter (Fig. 5*C* and *D*). The three parameters that the model includes are *L*, which describes the equilibrium between the open and closed states of the channel pore; $K_A$, which is the affinity of nucleotides for the inhibitory binding site of Kir6.2; and *D*, which represents the selective stabilization of particular conformations of the channel by nucleotide binding, such that $D < 1$ promotes closure of the channel and $D > 1$ promotes opening of the channel.

The fits to the MWC-type model resulted in parameter estimates for *L* that were indistinguishable between Kir6.2*+SUR1 and the three K39 mutants, suggesting that we were unable to detect any change in the open probability of the channel as a result of these substitutions. We observed reductions in the TNP-ATP binding affinity of each of the K39 mutants, although the posterior probability estimates of $K_A$ for Kir6.2*+SUR1 and Kir6.2*-K39R+SUR1 overlapped, meaning that this difference was not meaningful. In addition, fits to the Kir6.2*-K39E+SUR1 data yielded an estimate for *D* closer to one than that for Kir6.2*+SUR1, indicating a reduction in the selectivity of TNP-ATP for the closed state of the channel in this construct.

## Discussion

Our work suggests a mechanistic explanation for how PIP$_2$ and ATP binding to Kir6.2 influence one another to modulate K$_{ATP}$ channel activity. We show that the identity of the amino acid residue at position K39 is important for modulating the sensitivity of the channel to nucleotide inhibition, influencing both nucleotide binding affinity and the selectivity of nucleotides for the closed state.

Our atomistic MD simulations suggest that a single key residue, K39, forms part of both the ATP and PIP$_2$ binding sites on Kir6.2. When both ligands are present, K39 has a stronger preference for co-ordination with PIP$_2$ than with ATP (Fig. 2*C*). No other residue (in either site) alters its contact probability in the presence of the other ligand. This finding is in support of previous experimental work suggesting that ATP and PIP$_2$ effectively compete for binding to a given Kir6.2 subunit (Cukras et al., 2002; Enkvetchakul et al., 2001; MacGregor et al., 2002).

The simulation data further suggest that the K39R substitution in Kir6.2 that leads to transient neonatal diabetes (Zhang et al., 2015) might lead to a gain-of-function phenotype by increasing the strength of the interaction of the side-chain with PIP$_2$. *In vitro*, we would expect this to lead to an increase in PIP$_2$ affinity and a concomitant decrease in sensitivity of K$_{ATP}$ channel currents to inhibition by nucleotides. Indeed, we observed that introducing the K39R mutation into Kir6.2 increases the IC$_{50}$ for nucleotide inhibition by approximately 1.5- to 3-fold (Fig. 5*D*), supporting this finding.

Curiously, however, we observed that substitution to K39E (with a negatively charged side-chain) also leads to a reduction in sensitivity to nucleotide inhibition, and substitution to K39A (no charge) does not affect inhibition at all. These findings are at odds with our hypothesis on the importance of hydrogen bonding at this residue. Given that our simulation data suggest that K39 also co-ordinates ATP binding, we considered the possibility that we might be observing a mixture of effects on both ATP and $PIP_2$ regulation of the channel.

To disentangle these effects, we measured TNP-ATP binding to ANAP-labelled Kir6.2 constructs directly. Collection of binding data in parallel with current inhibition allows us to estimate the binding affinity of TNP-ATP for each of the Kir6.2*-K39 mutants (Fig. 5*D*). These experiments suggest that the K39E substitution reduces the binding affinity of nucleotides, in addition to any effects it might have on $PIP_2$ co-ordination, explaining why we do not observe a lower $IC_{50}$ for ATP inhibition of this construct. Unfortunately, although the $IC_{50}$ for ATP inhibition of Kir6.2*-K39A+SUR1 was indistinguishable from the $IC_{50}$ for ATP inhibition of Kir6.2*+SUR1

(Fig. 4*C*), we observed a noticeable difference between current inhibition of the two constructs by TNP-ATP, making it difficult to interpret our findings for this mutant.

In summary, we propose that residue K39 of Kir6.2 can co-ordinate either ATP or $PIP_2$ binding (but not both), in support of previous work (Fig. 7). Mutation of this residue to arginine results in neonatal diabetes, which we suggest occurs by enhancing the strength of the $PIP_2$ interaction with the side-chain, resulting in a loss of sensitivity to inhibition by nucleotides.

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

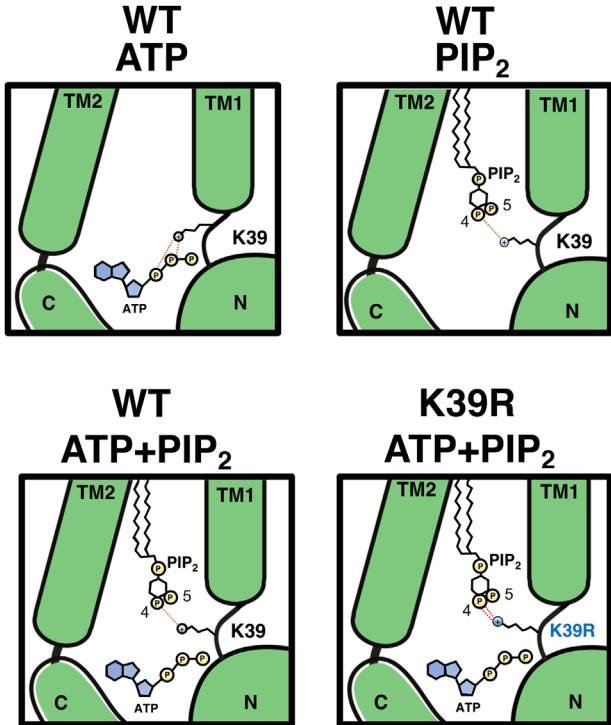

**Figure 7. Schematic representation of the interaction between Kir6.2, ATP and phosphatidylinositol 4,5-bisphosphate**
Interaction between ATP (silver sphere), phosphatidylinositol 4,5-bisphosphate [$PIP_2$; hexagon with silver phosphate group (P)] and Kir6.2 (green). A positive charge on K39 is denoted in yellow (or red in the K39R mutant). The orange dashed line represents hydrogen bonds between K39 and the ligand. The arrow represents the change in motion of K39 [Colour figure can be viewed at wileyonlinelibrary.com]

Enkvetchakul, D., Loussouarn, G., Makhina, E., & Nichols, C. G. (2001). ATP interaction with the open state of the KATP channel. *Biophysical Journal*, **80**(2), 719–728.

Enkvetchakul, D., Loussouarn, G., Makhina, E., Shyng, S. L., & Nichols, C. G. (2000). The kinetic and physical basis of K(ATP) channel gating: Toward a unified molecular understanding. *Biophysical Journal*, **78**(5), 2334–2348.

Enkvetchakul, D., & Nichols, C. G. (2003). Gating mechanism of KATP channels. *Journal of General Physiology*, **122**(5), 471 LP–480.

Fan, Z., & Makielski, J. C. (1999). Phosphoinositides decrease ATP sensitivity of the cardiac ATP-sensitive K(+) channel. A molecular probe for the mechanism of ATP-sensitive inhibition. *Journal of General Physiology*, **114**(2), 251–270.

De Franco, E., Saint-Martin, C., Brusgaard, K., Knight Johnson, A. E., Aguilar-Bryan, L., Bowman, P., Arnoux, J. B., Larsen, A. R., Sanyoura, M., Greeley, S. A. W., Calzada-León, R., Harman, B., Houghton, J. A. L., Nishimura-Meguro, E., Laver, T. W., Ellard, S., Del Gaudio, D., Christesen, H. T., Bellanné-Chantelot, C., & Flanagan, S. E. (2020). Update of variants identified in the pancreatic $\beta$-cell KATP channel genes KCNJ11 and ABCC8 in individuals with congenital hyperinsulinism and diabetes. *Human Mutation*, **41**(5), 884–905.

Gowers, R. J., Linke, M., Barnoud, J., Reddy, T. J. E., Melo, M. N., Seyler, S. L., Domański, J., Dotson, D. L., Buchoux, S., Kenney, I. M., & Beckstein, O. (2016). MDAnalysis: A python package for the rapid analysis of molecular dynamics simulations. *Proc 15th Python Sci Conf* 98–105.

Haider, S., Antcliff, J. F., Proks, P., Sansom, M. S. P., & Ashcroft, F. M. (2005). Focus on Kir6.2: A key component of the ATP-sensitive potassium channel. *Journal of Molecular and Cellular Cardiology*, **38**(6), 927–936.

Haider, S., Tarasov, A. I., Craig, T. J., Sansom, M. S. P., & Ashcroft, F. M. (2007). Identification of the PIP$_2$-binding site on Kir6.2 by molecular modelling and functional analysis. *EMBO Journal*, **26**(16), 3749–3759.

Hansen, S. B., Tao, X., & MacKinnon, R. (2011). Structural basis of PIP$^2$ activation of the classical inward rectifier K$^+$ channel Kir2.2. *Nature*, **477**(7365), 495–498.

Hilgemann, D. W., & Ball, R. (1996). Regulation of cardiac Na$^+$/Ca$^{2+}$ exchange and K$_{ATP}$ potassium channels by PIP$_2$. *Science*, **273**(5277), 956 LP–959.

Huang, J., Rauscher, S., Nawrocki, G., Ran, T., Feig, M., de Groot, B. L., Grubmüller, H., & MacKerell, A. D. (2017). CHARMM36m: An improved force field for folded and intrinsically disordered proteins. *Nature Methods*, **14**(1), 71–73.

Kim, S., Lee, J., Jo, S., Brooks III C. L., Lee H. S., & Im W. (2017). CHARMM-GUI ligand reader and modeler for CHARMM force field generation of small molecules. *Journal of Computational Chemistry*, **38**(21), 1879–1886.

Lee, K. P. K., Chen, J., & Mackinnon, R. (2017a). Molecular structure of human katp in complex with ATP and ADP. *Elife*, **6**, e32481.

Lee, K. P. K., Chen, J., & MacKinnon, R. (2017b). Molecular structure of human KATP in complex with ATP and ADP ed. Swartz KJ. *Elife*, **6**, e32481.

Li, N., Wu, J. X., Ding, D., Cheng, J., Gao, N., & Chen, L. (2017). Structure of a pancreatic ATP-sensitive potassium channel. *Cell*, **168**(1–2), 101–110.e10.

MacGregor, G. G., Dong, K., Vanoye, C. G., Tang, L., Giebisch, G., & Hebert, S. C. (2002). Nucleotides and phospholipids compete for binding to the C terminus of KATP channels. *Proceedings of the National Academy of Sciences of the United States of America*, **99**(5), 2726 LP–2731.

Männikkö, R., Stansfeld, P. J., Ashcroft, A. S., Hattersley, A. T., Sansom, M. S. P., Ellard, S., & Ashcroft, F. M. (2011). A conserved tryptophan at the membrane-water interface acts as a gatekeeper for Kir6.2/SUR1 channels and causes neonatal diabetes when mutated. *The Journal of Physiology*, **589**(13), 3071–3083.

Marrink, S. J., Risselada, H. J., Yefimov, S., Tieleman, D. P., & de Vries, A. H. (2007). The MARTINI force field: Coarse grained model for biomolecular simulations. *Journal of Physical Chemistry B*, **111**(27), 7812–7824.

Martin, G. M., Kandasamy, B., DiMaio, F., Yoshioka, C., & Shyng, S.-L. (2017). Anti-diabetic drug binding site in a mammalian KATP channel revealed by Cryo-EM ed. Swartz KJ. *Elife*, **6**, e31054.

Martin, G. M., Sung, M. W., Yang, Z., Innes, L. M., Kandasamy, B., David, L. L., Yoshioka, C., & Shyng, S.-L. (2019). Mechanism of pharmacochaperoning in a mammalian KATP channel revealed by cryo-EM ed. Aldrich R, Yellen G, Moiseenkova-Bell VY, Nichols CG & Agar J. *Elife*, **8**, e46417.

Martin, G. M., Yoshioka, C., Rex, E. A., Fay, J. F., Xie, Q., Whorton, M. R., Chen, J. Z., & Shyng, S. L. (2017). Cryo-EM structure of the ATP-sensitive potassium channel illuminates mechanisms of assembly and gating. *eLife*, **6**, e24149.

Michaud-Agrawal, N., Denning, E. J., Woolf, T. B., & Beckstein, O. (2011). MDAnalysis: A toolkit for the analysis of molecular dynamics simulations. *Journal of Computational Chemistry*, **32**(10), 2319–2327.

Monticelli, L., Kandasamy, S. K., Periole, X., Larson, R. G., Tieleman, D. P., & Marrink, S.-J. (2008). The MARTINI coarse-grained force field: Extension to proteins. *Journal of Chemical Theory and Computation*, **4**(5), 819.

Olesen, K., Awasthi, N., Bruhn, D. S., Pezeshkian, W., & Khandelia, H. (2018). Faster simulations with a 5 fs time step for lipids in the CHARMM force field. *Journal of Chemical Theory and Computation*, **14**(6), 3342–3350.

Parrinello, M., & Rahman, A. (1981). Polymorphic transitions in single crystals: A new molecular dynamics method. *Journal of Applied Physics*, **52**(12), 7182–7190.

Pipatpolkai, T., Corey, R. A., Proks, P., Ashcroft, F. M., & Stansfeld, P. J. (2020). Evaluating inositol phospholipid interactions with inward rectifier potassium channels and characterising their role in disease. *Communications Chemistry*, **3**(1), 147.

Pipatpolkai, T., Usher, S., Stansfeld, P. J., & Ashcroft, F. M. (2020). New insights into KATP channel gene mutations and neonatal diabetes mellitus. *Nature Reviews Endocrinology*, **16**(7), 378–393.

Proks, P., de Wet, H., & Ashcroft, F. M. (2010). Activation of the K$_{ATP}$ channel by Mg-nucleotide interaction with SUR1. *Journal of General Physiology*, **136**(4), 389–405.

Puljung, M., Vedovato, N., Usher, S., & Ashcroft, F. (2019). Activation mechanism of ATP-sensitive K$^+$ channels explored with real-time nucleotide binding. *eLife*, **8**, e41103.

Puljung, M. C. (2018). Cryo-electron microscopy structures and progress toward a dynamic understanding of KATP channels. *Journal of General Physiology*, **150**(5), 653–669.

Rafiq, M., Flanagan, S. E., Patch, A.-M., Shields, B. M., Ellard, S., & Hattersley, A. T. (2008). Effective treatment with oral sulfonylureas in patients with diabetes due to sulfonylurea receptor 1 (SUR1) mutations. *Diabetes Care*, **31**(2), 204 LP–209.

Rorsman, P., & Ashcroft, F. M. (2018). Pancreatic $\beta$-cell electrical activity and insulin secretion: Of mice and men. *Physiological Reviews*, **98**(1), 117–214.

Schrodinger, L. L. C. (2015). The PyMOL Molecular Graphics System, Version 1.8.

Shyng, S.-L., Cukras, C. A., Harwood, J., & Nichols, C. G. (2000). Structural determinants of PIP$_2$ regulation of inward rectifier K$_{ATP}$ channels. *Journal of General Physiology*, **116**(5), 599 LP–608.

Shyng, S. L., & Nichols, C. G. (1998). Membrane phospholipid control of nucleotide sensitivity of K$_{ATP}$ channels. *Science*, **282**(5391), 1138–1141.

Van Der Spoel, D., Lindahl, E., Hess, B., Groenhof, G., Mark, A. E., & Berendsen, H. J. C. (2005). GROMACS: Fast, flexible, and free. *Journal of Computational Chemistry*, **26**(16), 1701–1718.

Stansfeld, P. J., Goose, J. E., Caffrey, M., Carpenter, E. P., Parker, J. L., Newstead, S., & Sansom, M. S. (2015). MemProtMD: Automated insertion of membrane protein structures into explicit lipid membranes. *Struct England 1993*, **23**, 1350–1361.

Stansfeld, P. J., Hopkinson, R., Ashcroft, F. M., & Sansom, M. S. P. (2009). PIP2-binding site in Kir channels: Definition by multiscale biomolecular simulations. *Biochemistry*, **48**(46), 10926–10933.

Stansfeld, P. J., & Sansom, M. S. P. (2011). From coarse grained to atomistic: A serial multiscale approach to membrane protein simulations. *Journal of Chemical Theory and Computation*, **7**(4), 1157–1166.

Theobald, D. L. (2005). Rapid calculation of RMSDs using a quaternion-based characteristic polynomial. *Acta Crystallographica. Section A, Foundations of crystallography*, **61**(4), 478–480.

Trapp, S., Proks, P., Tucker, S. J., & Ashcroft, F. M. (1998). Molecular analysis of ATP-sensitive K channel gating and implications for channel inhibition by ATP. *Journal of General Physiology*, **112**, 333 LP–349.

Trapp, S., Tucker, S. J., & Ashcroft, F. M. (1997). Activation and inhibition of K-ATP currents by guanine nucleotides is mediated by different channel subunits. *Proceedings of the National Academy of Sciences of the United States of America*, **94**(16), 8872 LP–8877.

Tucker, S. J., Gribble, F. M., Proks, P., Trapp, S., Ryder, T. J., Haug, T., Reimann, F., & Ashcroft, F. M. (1998). Molecular determinants of KATP channel inhibition by ATP. *EMBO Journal*, **17**(12), 3290–3296.

Tucker, S. J., Gribble, F. M., Zhao, C., Trapp, S., & Ashcroft, F. M. (1997). Truncation of Kir6.2 produces ATP-sensitive K+ channels in the absence of the sulphonylurea receptor. *Nature*, **387**(6629), 179–183.

Usher, S. G., Ashcroft, F. M., & Puljung, M. C. (2020). Nucleotide inhibition of the pancreatic ATP-sensitive K+ channel explored with patch-clamp fluorometry. *eLife*, **9**, e52775.

Whorton, M. R., & MacKinnon, R. (2011). Crystal structure of the mammalian GIRK2 K+ channel and gating regulation by G proteins, PIP2, and sodium. *Cell*, **147**(1), 199–208.

Wu, J.-X., Ding, D., Wang, M., Kang, Y., Zeng, X., & Chen, L. (2018). Ligand binding and conformational changes of SUR1 subunit in pancreatic ATP-sensitive potassium channels. *Protein Cell*, **9**(6), 553–567.

Yan, F.-F., Lin, Y.-W., MacMullen, C., Ganguly, A., Stanley, C. A., & Shyng, S.-L. (2007). Congenital hyperinsulinism associated ABCC8 mutations that cause defective trafficking of ATP-sensitive K+ channels: Identification and rescue. *Diabetes*, **56**(9), 2339–2348.

Zhang, M., Chen, X., Shen, S., Li, T., Chen, L., Hu, M., Cao, L., Cheng, R., Zhao, Z., & Luo, F. (2015). Sulfonylurea in the treatment of neonatal diabetes mellitus children with heterogeneous genetic backgrounds. *Journal of Pediatric Endocrinology & Metabolism*, **28**(7–8), 877–884.

Zhao, C., & MacKinnon, R. (2021). Molecular structure of an open human K$_{ATP}$ channel. *Proceedings of the National Academy of Sciences of the United States of America*, **118**, e2112267118.

## Additional information

### Data availability statement

All data in a non-identifying format are securely stored at the Universities of Oxford and Warwick, UK. The data are available upon request.

### Competing interests

The authors declare no competing interests.

### Author contributions

T.P. performed the coarse-grained and atomistic molecular dynamics simulations. N.V. and S.U. performed the molecular biology, electrophysiology and fluorescence measurements. All authors jointly designed the experiments, analysed the data and wrote the manuscript. All authors have read and approved the final version of this manuscript and agree to be accountable for all aspects of the work in ensuring that questions related to the accuracy or integrity of any part of the work are appropriately investigated and resolved. All persons designated as authors qualify for authorship, and all those who qualify for authorship are listed.

## Funding

T.P. and S.U. hold a Wellcome Trust OXION studentship. T.P. holds a Clarendon scholarship. Research in P.J.S.'s laboratory is funded by Wellcome (208361/Z/17/Z), the MRC (MR/S009213/1) and BBSRC (BB/P01948X/1, BB/R002517/1 and BB/S003339/1). Research in F.M.A.'s laboratory is funded by the MRC (MR/T002107/1) and BBSRC (BB/R002517/1, BB/R017220/1). This project made use of time on ARCHER and JADE granted via the UK High-End Computing Consortium for Biomolecular Simulation, HECBioSim (http://hecbiosim.ac.uk), supported by EPSRC (grant no. EP/R029407/1).

## Acknowledgements

The authors thank Michael C. Puljung for discussions about the MWC modelling and interpretations of the parameters. The authors also thank Irfan Alibay, Robin Corey, Michael Horrell, Raul Terron-Exposito and Owen Vickery for advice and technical support.

## Keywords

ATP-sensitive potassium channel, molecular dynamics, phosphatidylinositol 4,5-bisphosphate

## Supporting information

Additional supporting information can be found online in the Supporting Information section at the end of the HTML view of the article. Supporting information files available:

**Statistical Summary Document**
**Peer Review History**

