## [Peer Review History · The Journal of Physiology]

The dynamic interplay of PIP2 and ATP in the regulation of the KATP channel

Tanadet Pipatpolkai, Samuel G Usher, Natascia Vedovato, Frances M. Ashcroft, and Phillip Stansfeld
DOI: 10.1113/JP283345

Corresponding author(s): Phillip Stansfeld (phillip.stansfeld@warwick.ac.uk)

The following individual(s) involved in review of this submission have agreed to reveal their identity: Anna Stry-Weinzinger (Referee #2); Catherine Vénien-Bryan (Referee #3); Colin G Nichols (Referee #4)

Review Timeline:

Submission Date:	04-Feb-2022
Editorial Decision:	16-Mar-2022
Resubmission Received:	19-May-2022
Accepted:	04-Aug-2022

Senior Editor: Ian Forsythe

Reviewing Editor: Thomas DeCoursey

Transaction Report:

Dear Dr Stansfeld,

Re: JP-RP-2022-282938 "The dynamic interplay of PIP2 and ATP in the regulation of the KATP channel" by Tanadet Pipatpolkai, Samuel G Usher, Natascia Vedovato, Frances M. Ashcroft, and Phillip Stansfeld

Thank you for submitting your manuscript to The Journal of Physiology. It has been assessed by a Reviewing Editor and by 4 Referees, including a statistics editor, and the reports are copied below.

Please let your co-authors know of the following editorial decision as quickly as possible.

As you will see, in its current form, the manuscript is not acceptable for publication in The Journal of Physiology. In comments to me, the Reviewing Editor expressed interest in the potential of this study, but much work still needs to be done (and this may include new experiments) in order to satisfactorily address the concerns raised in the reports.

In view of this interest, I would like to offer you the opportunity to carry out all of the changes requested in full, and to resubmit a new manuscript using the "Submit Special Case Resubmission for JP-RP-2022-282938..." on your homepage.

We cannot, of course, guarantee ultimate acceptance at this stage as the revisions required are substantial. However, we encourage you to consider the requested changes and resubmit your work to us if you are able to complete or address all changes.

A new manuscript would be renumbered and redated, but the original referees would be consulted wherever possible. An additional referee's opinion could be sought, if the Reviewing Editor felt it necessary. A full response to each of the reports should be uploaded with a new version.

I hope that the points raised in the reports will be helpful to you.

Yours sincerely,

Ian D. Forsythe
Deputy Editor-in-Chief
The Journal of Physiology
<https://jp.msubmit.net>
<http://jp.physoc.org>
The Physiological Society
Hodgkin Huxley House
30 Farringdon Lane
London, EC1R 3AW
UK
<http://www.physoc.org>
<http://journals.physoc.org>

EDITOR COMMENTS

Reviewing Editor:

Three reviewers have evaluated this manuscript. Although they all find substantial merit, they have identified a number of problems that limit their enthusiasm. Some of these are potentially fatal flaws.

1) Overall, the advances arising from this study appear somewhat incremental, despite the clear importance of the subject in general (Rev. #2: the study provides somewhat limited insight - with the key finding being that a single amino acid, namely K39 interacts with both ATP and PIP2").

2) A large component is MD simulations. Reviewer #2 finds the approach taken to be less than optimal, and that it failed to incorporate evidence from other studies. Reviewer #4 points out a number of specific areas in which the MD could not be matched with experimental data.

3) Reviewers #3 and #4 find the FRET data difficult to interpret and irreconcilable with their model predictions.

4) Reviewer #4 points out in detail a number of substantial misinterpretations of previous studies. These cast doubt on both the novelty and the validity of the present results.

5) Reviewers #2 and #4 point out recent cryo-EM studies need to be reconciled with the model proposed here.

6) Reviewer #4 lists a number of serious problems that must be addressed before the manuscript could be published. Most alarming is that this paper is identical to one submitted to another journal, and the authors have resubmitted without correcting any of these flaws.

Senior Editor:

Although significant issues have been raised by the referees, if the authors are prepared to fully address these, then a resubmission of this article would be encouraged.

REFEREE COMMENTS

Referee #1:

I have been asked for an initial opinion on the statistics described in this paper. I have reviewed the specified models and find this analysis to be robust, with reasonable prior distributions specified for each parameter and posterior distributions presented appropriately. I note also that the authors discussed the models and interpretation of the parameters from the MWC modelling with Michael C. Puljung, a recognised expert in this area, which provides additional assurance.

Referee #2:

In this manuscript, Tanadet Pipatpolkai et al. have performed molecular dynamics (MD) simulation, electrophysiology and fluorescence spectroscopy to investigate the dynamic interplay of PIP2 and ATP in the regulation of the KATP channel.

While the paper investigates an important question and constitutes original research, there are several key points that need to be clarified:

Study design

Why did the authors start from a coarse-grained simulation setup, while running production runs for atomistic simulations? I am particular confused about the following sentence: "The position of the coarse-grained PIP2 is taken from the chicken Kir2.2-PIP2:diC8 after conversion to a

coarse grain model (Hansen et al., 2011). Wouldn't it have been easier to use the original crystal structure information from Mackinnon's lab, which already contains atomistic information on the position of a short-chain PIP2?

The position of PIP2 is also available for Kir3.2. Given the crucial importance of placing PIP2, did the authors compare/consider this information as well? Is PIP2 binding in a similar fashion in these different KIR channels?

The ATP-binding site on Kir6.2 has been identified in several cryo-EM structures. What was the rationale to use 6BAA for simulations only? It has been shown that ATP adopts different rotamers at the γ phosphate in different structures (e.g. 6BAA vs. 6C30).

Lee et al, report in their paper that in the ATP bound KATP structures (propeller and quatrefoil) the PIP2 binding site is substantially compressed - isn't this also the case in the 6BAA structure? How did this affect PIP2 placement in Kir6?

Results section:

Q52 contacts PIP2 in their simulation, a residue that has been shown crucial for KIR-SUR coupling previously. Maybe a more thorough discussion on the limitations of simulating the pore only should be included.

The authors simulated different systems (apo/PIP2/ATP/PIP2+ATP). Do they see any conformational changes in the channel, e.g. changes at the C-linker, CTD etc., or are the simulations too short to see such changes?

A very recent cryo-EM paper by the Mackinnon group reports an open KATP conformation, associated with coordinated structural changes within the ATP binding site, independent of PIP2. How do these findings align with the current predictions?

Figure 5A: why was ATP applied for different time length in the different constructs?

Discussion:

Overall, the discussion is rather short and could be improved, particularly with respect to better discussing the findings with respect to the current literature in the field.

Minor:

Figure 1 inset: K69 - should be labeled K67?, same in Fig. 3A

Referee #3:

In this paper entitled « The dynamic interplay of PIP2 and ATP in the regulation of the KATP channel » Pipatpolkai et al. employed a combination of MD simulation, electrophysiology and fluorescence (Voltage Clamp Fluorimetry) to investigate the role of a crucial amino acid, K39, in the binding of PIP2 and ATP. Importantly, the mutation K39R causes neo-natal diabetes. Influence of this mutation on the ATP and PIP2 binding was investigated.

ATP-sensitive potassium (KATP) channels couple the intracellular ATP concentration to insulin secretion. This channel is inhibited by ATP binding to the Kir6.2 tetramer and activated by PIP2. ATP and PIP2 effectively compete for binding to a given Kir6.2 subunit. The MD simulation shows how K39 interacts with PIP2 and ATP and when both ligands are present, K39 has a stronger preference for co-ordination with PIP2 than with ATP. ATP occupancy, PIP2 occupancy experiments are well described.

K39R leads to transient neonatal diabetes. In this paper, MD simulation proves that K39R increases the strength of the interaction of residue to PIP2. These data explain clearly the mechanism by which the mutation impairs insulin secretion and leads to neonatal diabetes. More over electrophysiology experiments shows that the introduction of K39R mutation into Kir6.2 increases the IC50 for nucleotide inhibition by about 1.5-3 fold.

PCF was then performed on ANAP-labeled Kir6.2* construct (on W311) in order to estimate the binding affinity of TNP-ATP for each of the Kir6.2*-K39 mutants (A,E and R) using FRET experiments coupled to electrophysiology. However the results are difficult to interpret. Fits of the MWC-type model shows that it was not possible to distinguish any change in the open probability of the channel.

These data are very interesting, the data are strong.

I have minor comments

Figure 1 : K69 should be P69

Figure 5 : Concentration of ATP are given in M (Should be added)

Figure 7 : WT-ATP shows that K39 interacts with alpha and beta P of the ATP. This is in contradiction with the text line 14 page 6 : "K39 interacts with gamma P " Should be corrected In the legend it is mentioned an arrow « The arrow represents the change in motion of K39". Where is the arrow on the figure?

The PCF experiments : Could you explain why the W331 was chosen for the position of the ANAP. A supplementary figure of the structure of Kir6.2 with the locations of W311 (ANAP) and the TNP-ATP should be provided along with the location of the R39 mutants. This will be useful to understand the FRET experiments.

Referee #4:

NB Authors will see that the review below was written for previous submission of this manuscript to a different journal, but comparison of the two manuscripts indicates they are identical, so comments below should all be addressed in any revision.

General:

The authors have used computational and experimental analyses to assess the interplay between ATP inhibition and PIP2 activation of KATP channels. This is a key molecular area of regulation of these channels and one that has been extensively studied previously. The authors bring some novel approaches to the issue, but there are concerns with the approach and interpretation, as detailed in the comments below.

Major:

1. A major concern regarding the modeling is that the results are described as facts, rather than testable predictions. This is a fundamental issue that needs to be acknowledged.

2. Experimentally, there is major concern that, as the paper mentioned, there are inconsistencies between ATP inhibition and TNP-ATP inhibition. The main potential novelty of the modeling lies in identifying K39 as a residue that can interact with phosphate groups of both ATP and PIP2. The authors acknowledge (Fig. S7) that the TNP group may interfere with the binding at K39, but the FRET assay and model are both for TNP-ATP which may not be able to explain the mechanism of ATP inhibition of the current.

3. The authors used MD simulations and experimental fluoro-patching to see how certain mutations affect ATP binding and channel activity, from which molecular mechanisms of apparent exclusive binding of PIP2 and ATP to KATP proteins may be inferred. Based on their MD simulations, the authors claim K39 is the key residue to facilitate PIP2 binding over ATP and that increased hydrogen bonds to PIP2 explains gain of function of the K39R mutant. Experimentally they tried to correlate the changes in ATP binding and ATP inhibition of several mutant channels, although this was not very successful. First, the IC50 for TNP-ATP current inhibition and EC50 of TNP-ATP FRET differed by 2 orders of magnitude, making it impossible to directly correlate TNP-ATP binding to functional modulation. Second, the functional results regarding K39 mutants are not in accordance with their simulation results: simulations imply that hydrogen bonding between the K39 amine and PIP2 is critical for PIP2 binding. If this is correct, then K39A and K39E should give a loss of function phenotype, which is not observed. K39E shows a gain of function phenotype, which is even stronger in the absence of SUR1, best mimicking the MD simulations. This questions whether K39R gain of function effect can be attributed to increased interaction with PIP2? Could it be a gating mutant with increased open state stability independent of PIP2 binding?

4. There seems to be confusion regarding previous studies that have analyzed the interaction of ATP and PIP2 in regulating the channel (p.3 last paragraph). The authors say "Because an increased channel open probability is associated with reduced ATP inhibition^{20,21}, it is possible that at least part of the effect of PIP2 is mediated via changes in Popen. However, it has also been argued that PIP2 may have an additional effect on ATP sensitivity that is independent of Popen²⁰". On p6 they say "Previous studies have shown that PIP2 reduces channel ATP inhibition^{4,19,30,31}. However, it was not clear if PIP2 competes directly for the ATP binding site or if it interferes with ATP dependent gating (or both)." These statements misrepresent previous studies, particularly multiple studies by Enkvetchakul/Nichols/Shyng groups [including ref 20, 30, and unreferenced Biophys J. 2001;80(2):719-28; J Gen Physiol. 2003 Nov;122(5):471-80]. Those studies mechanistically explain how the change in Popen resulting from PIP2 binding causes loss of ATP sensitivity, and do not suggest any additional effect on ATP sensitivity: the first paper to report PIP2 modulation of both Popen and ATP was ref 30, which specifically discussed how 'negative heterotropic cooperativity' between the two ligands (meaning that they both compete for the same unliganded channel, without having to bind at the same site) could explain the effect of PIP2 on ATP sensitivity as a direct consequence of the effect on Popen, as was subsequently quantitatively confirmed and modeled by Enkvetchakul et al.

5. The authors go on to say "previous studies have proposed that PIP2 competes with ATP for the same binding site on the C-terminus of the protein²². However, comparison of recent structural studies of the channel with bound ATP^{5,6}, and docking and molecular dynamics simulations with PIP2 suggest that ATP..." Although the authors quote ref 22 as having proposed that PIP2 and ATP compete for the same binding site, that study was carried out on Kir1.1, which is not a KATP channel. Since the Enkvetchakul studies show how 'competition' between PIP2 and ATP arises without the two ligands binding in the same pocket, the point being made about the binding pockets being different is not an argument against competition. The authors describe what they consider to be two alternate concepts for how PIP2 affects ATP sensitivity, but they are really the same - PIP2 competes with ATP for binding to the unliganded channel, what the authors describe as a "local allosteric effect" is the same as 'negative heterotropic cooperativity'. Even though PIP2 and ATP may not bind at the same site (the sites could be far apart), ATP binding will still be reduced if PIP2 is increased, because the fraction of unliganded channels will be reduced.

6. How valid or reliable is the PIP2 bound structure, derived from CG-MD? How can this be reconciled with the fact that Lys170 at the bottom of the TM2 and E179, both critical for PIP2 gating, are not directly interacting with PIP2 at all in the simulations? Also, the mode of PIP2 binding to Kir channels is quite different from what has been observed in many crystal structures where the 5' P makes more extensive interactions with the neighboring basic residues while the 4' P makes limited interactions and faces away from the protein?

7. Why was E179 not analyzed in MD trajectories in the same way as done for K39? The gain of function effect of E179A or E179K is even greater than K39R, and the kinetic model suggests that E179A and E179K also reduce the nucleotide binding affinity.

Other comments

4. Fig1A residue labeling is incorrect: R54 and R176 should be switched, and the text describing that R54 and K67 from one subunit and R176 from an adjacent subunit is incorrect.

5. K67 and R176 are from the same subunit and R54 is from neighboring subunit.

What is the basis for considering RMSF > 1Angstrom to be biologically significant?

6. The amine group of Lys residues is better described as a terminal amine group rather than a head group. A head group is the term used to describe lipid structures, and it is unconventional to call an aa side chain part a head group.

7. The finding that K39 may interact with either ATP or PIP2 is very interesting and suggests it may actually be involved in both binding sites. However, cryo-EM structures have only shown CTD-disengaged conformations for Kir6.2 (as opposed to the 'engaged' conformations that are also seen in Kir2.2 and Kir3.2 structures and which likely represent the active conformations), which results in a quite a distance between the PIP2 and ATP binding site of about 25 Å as mentioned in the manuscript. Presumably the simulations involve 'engagement' and the binding sites are not that far apart in these simulations? Showing the distance between the two sites in their simulations will help the reader to understand how one of residue can interact with both substrates.

8. Fig 2A R54 is mislabeled and it is likely to be R176.

9. Would the increased H-bonds between K39R and PIP2 also increase H-bonds between K39R and ATP? If the mutation increased interaction for both substrates, this would argue against the gain of function phenotype of the K39R mutant being

due to increased PIP2 affinity. Therefore, it is necessary to show how the H-bonding pattern differs for K39 versus K39R and ATP. The postulate, 'K39R will lead to reduced channel inhibition by ATP, and thereby impairs insulin secretion leading to neonatal diabetes' is questionable.

10. In the discussion; the authors suggest that steric clashes may interfere with TNP-ATP binding to K39R mutant. It is difficult to imagine that the K to R mutation could create much difference and, in addition, K39R shows yet stronger binding ($IC_{50} = 2.62 \mu M$) to the site than K39A ($IC_{50} = 13.2 \mu M$). If steric occlusion was the cause, this should be relieved with the smaller sidechain in the K39A mutation and the binding should then be stronger?

11. Ref 14 and 27 are the same.

ADDITIONAL FORMATTING REQUIREMENTS:

-Include a Key Points list in the article itself, before the Abstract.

-Author photo and profile. First (or joint first) authors are asked to provide a short biography (no more than 100 words for one author or 150 words in total for joint first authors) and a portrait photograph. These should be uploaded and clearly labelled with the revised version of the manuscript. See Information for Authors for further details.

-The Journal of Physiology funds authors of provisionally accepted papers to use the premium BioRender site to create high resolution schematic figures. Follow this link and enter your details and the manuscript number to create and download figures. Upload these as the figure files for your revised submission. If you choose not to take up this offer we require figures to be of similar quality and resolution. If you are opting out of this service to authors, state this in the Comments section on the Detailed Information page of the submission form.

-Please upload separate high-quality figure files via the submission form.

-Please ensure that the Article File you upload is a Word file.

-A Statistical Summary Document, summarising the statistics presented in the manuscript, is required upon revision. It must be on the Journal's template, which can be downloaded from the link in the Statistical Summary Document section here: https://jp.msubmit.net/cgi-bin/main.plex?form_type=display_requirements#statistics

-Papers must comply with the Statistics Policy https://jp.msubmit.net/cgi-bin/main.plex?form_type=display_requirements#statistics

In summary:

-If $n \leq 30$, all data points must be plotted in the figure in a way that reveals their range and distribution. A bar graph with data points overlaid, a box and whisker plot or a violin plot (preferably with data points included) are acceptable formats.

-If $n > 30$, then the entire raw dataset must be made available either as supporting information, or hosted on a not-for-profit repository e.g. FigShare, with access details provided in the manuscript.

- n clearly defined (e.g. x cells from y slices in z animals) in the Methods. Authors should be mindful of pseudoreplication.

-All relevant n values must be clearly stated in the main text, figures and tables, and the Statistical Summary Document (required upon revision)

-The most appropriate summary statistic (e.g. mean or median and standard deviation) must be used. Standard Error of the Mean (SEM) alone is not permitted.

-Exact p values must be stated. Authors must not use 'greater than' or 'less than'. Exact p values must be stated to three significant figures even when 'no statistical significance' is claimed.

-Statistics Summary Document completed appropriately upon revision

-A Data Availability Statement is required for all papers reporting original data. This must be in the Additional Information section of the manuscript itself. It must have the paragraph heading "Data Availability Statement". All data supporting the results in the paper must be either: in the paper itself; uploaded as Supporting Information for Online Publication; or archived in an appropriate public repository. The statement needs to describe the availability or the absence of shared data. Authors must include in their Statement: a link to the repository they have used, or a statement that it is available as Supporting Information; reference the data in the appropriate sections(s) of their manuscript; and cite the data they have shared in the References section. Whenever possible the scripts and other artefacts used to generate the analyses presented in the paper should also be publicly archived. If sharing data compromises ethical standards or legal requirements then authors are not expected to share it, but must note this in their Statement. For more information, see our Statistics Policy.

-Please include an Abstract Figure. The Abstract Figure is a piece of artwork designed to give readers an immediate understanding of the research and should summarise the main conclusions. If possible, the image should be easily 'readable' from left to right or top to bottom. It should show the physiological relevance of the manuscript so readers can assess the importance and content of its findings. Abstract Figures should not merely recapitulate other figures in the manuscript. Please try to keep the diagram as simple as possible and without superfluous information that may distract from the main conclusion(s). Abstract Figures must be provided by authors no later than the revised manuscript stage and should be uploaded as a separate file during online submission labelled as File Type 'Abstract Figure'. Please ensure that you include the figure legend in the main article file. All Abstract Figures should be created using BioRender. Authors should use The Journal's premium BioRender account to export high-resolution images. Details on how to use and access the premium account are included as part of this email.

Confidential Review

04-Feb-2022

Referee #1:

I have been asked for an initial opinion on the statistics described in this paper. I have reviewed the specified models and find this analysis to be robust, with reasonable prior distributions specified for each parameter and posterior distributions presented appropriately. I note also that the authors discussed the models and interpretation of the parameters from the MWC modelling with Michael C. Puljung, a recognised expert in this area, which provides additional assurance.

We thank reviewer for the positive feedback and reassurance on our fitting protocol.

Referee #2:

In this manuscript, Tanadet Pipatpolkai et al. have performed molecular dynamics (MD) simulation, electrophysiology and fluorescence spectroscopy to investigate the dynamic interplay of PIP₂ and ATP in the regulation of the KATP channel.

While the paper investigates an important question and constitutes original research, there are several key points that need to be clarified:

Study design

Why did the authors start from a coarse-grained simulation setup, while running production runs for atomistic simulations? I am particular confused about the following sentence: "The position of the coarse-grained PIP₂ is taken from the chicken Kir2.2-PIP₂:diC8 after conversion to a coarse grain model (Hansen et al., 2011). Wouldn't it have been easier to use the original crystal structure information from Mackinnon's lab, which already contains atomistic information on the position of a short-chain PIP₂?"

The position of PIP₂ is also available for Kir3.2. Given the crucial importance of placing PIP₂, did the authors compare/consider this information as well? Is PIP₂ binding in a similar fashion in these different KIR channels?

We thank reviewer for the comments. We converted PIP₂ from di-C8 PIP₂ in the chicken structure to the CG representation to equilibrate the binding pose to fit with the Kir6.2, then re-convert the structure back to atomistic, and then further equilibrate the binding pose of PIP₂ within the binding site for an additional 80 ns. We conduct those multi-step conversions to simply adjust the binding pose of PIP₂ to fit with the Kir2.2. To confirm our binding site, we have aligned the position of the inositol group from diC8:PIP₂ chicken Kir2.2 structure and relax the position of PIP₂ using similar equilibration protocols (5 ns C_α restrained, PIP₂ restrained + 15 ns C_α restrained. All restraint were acting on xyz co-ordinates at 1000 kJ/nm²/mol). We compared the position of PIP₂ during the final 15 ns of our equilibration where the position of PIP₂ is unrestrained, with the final snapshot from our 380ns simulation (Rebuttal figure 1). This alignment shows that our binding site is within the conformational space of PIP₂ obtained from the crystal structure and thus, validated our binding site.

Rebuttal figure 1: Conformational sampling of PIP₂ during the position restrained run

The position of PIP₂ during the 15 ns are shown as gradient from blue (t = 0 ns) to red (t = 15 ns). The initial position of PIP₂ headgroup (blue) was derived from the chicken Kir2.2 crystal structure (PDB ID: 3SPI). The final position of PIP₂ and Kir6.2 obtained from our 380 ns simulation is shown in orange.

The ATP-binding site on Kir6.2 has been identified in several cryo-EM structures. What was the rationale to use 6BAA for simulations only? It has been shown that ATP adopts different rotamers at the γ phosphate in different structures (e.g. 6BAA vs. 6C30).

Lee et al, report in their paper that in the ATP bound KATP structures (propeller and quatrefoil) the PIP₂ binding site is substantially compressed - isn't this also the case in the 6BAA structure? How did this affect PIP₂ placement in Kir6?

We thank reviewer for the suggestion. We have now conducted additional simulations (3 repeats x 380 ns) with both PIP₂ and ATP bound in the quatrefoil structure (6C30). Here, we observed that there is no significance difference between the contact between K39 and ATP between quatrefoil and the propeller conformation (Supplementary figure 4). We have also highlighted that in the quatrefoil conformation, the contact between K39 in an absence of PIP₂ is greater than when the PIP₂ molecule is present. These information are now shown in figure 4.

Results section:

Q52 contacts PIP₂ in their simulation, a residue that has been shown crucial for KIR-SUR coupling previously. Maybe a more thorough discussion on the limitations of simulating the pore only should be included.

We thank reviewer for the comments suggesting the simulation of the full complex. The simulation box where four SUR1 subunits are present is at the size of 1.2 million atoms in each simulation box, whereas simulation box with just Kir6.2 is containing approximately 230k atoms. Nevertheless, we aimed to understand the role of SUR1 in this situation. We have now added an additional of 100 ns of the full K_{ATP} complex (6BAA) simulation to highlight an importance of Q52. Unfortunately, we do not observe any contact made between Q52 and PIP₂ during our simulation.

The authors simulated different systems (apo/PIP₂/ATP/PIP₂+ATP). Do they see any conformational changes in the channel, e.g. changes at the C-linker, CTD etc., or are the simulations too short to see such changes?

We thank reviewer for the additional question. However, it is very unlikely for us to observe any key conformational change in the CTD or C-linker under short (380ns) simulation timescale. It could be dangerous to interpolate such conformational change from our data. Recent study from Bründl et al. highlight that conformational change on Kir6.2 channel in the presence of PIP₂ is very little even under 1 μ s timescale.

A very recent cryo-EM paper by the Mackinnon group reports an open KATP conformation, associated with coordinated structural changes within the ATP binding site, independent of PIP₂. How do these findings align with the current predictions?

We thank reviewer for the suggestion. We have now conduct additional simulations (3 repeats) with PIP₂ bound in their open state conformation. Here, we observed similar contact profile between PIP₂ and the Kir6.2 even in the mutant channel. We also observed additional contacts between Kir6.2 and the hydrophobic residues, L72 and V151 which are not observed in any other conformations.

Figure 5A: why was ATP applied for different time length in the different constructs?

We thank reviewer for the concern on the time length. However, brief applications of ATP are commonly used in these types of experiments. In our studies, we applied ATP roughly for 5-10s. Sometimes, we will be using longer applications if the current wouldn't have

reached a steady state within the first 10 seconds. Long application of ATP is not recommended due to run-down property of the channel.

Discussion:

Overall, the discussion is rather short and could be improved, particularly with respect to better discussing the findings with respect to the current literature in the field.

We thank the reviewer for the suggestion. We have now increased the length of the discussion and update recent finding (MacKinnon, Bründl, Shyng) – with link to Kir6.1 channel.

Minor:

Figure 1 inset: K69 - should be labeled K67?, same in Fig. 3A

We thank the reviewer for the comment. We have now fixed the typo in the figure.

Referee #3:

In this paper entitled « The dynamic interplay of PIP2 and ATP in the regulation of the KATP channel » Pipatpolkai et al. employed a combination of MD simulation, electrophysiology and fluorescence (Voltage Clamp Fluorimetry) to investigate the role of a crucial amino acid, K39, in the binding of PIP2 and ATP. Importantly, the mutation K39R causes neo-natal diabetes . Influence of this mutation on the ATP and PIP2 binding was investigated.

ATP-sensitive potassium (KATP) channels couple the intracellular ATP concentration to insulin secretion. This channel is inhibited by ATP binding to the Kir6.2 tetramer and activated by PIP2. ATP and PIP2 effectively compete for binding to a given Kir6.2 subunit. The MD simulation shows how K39 interacts with PIP2 and ATP and when both ligands are present, K39 has a stronger preference for co-ordination with PIP2 than with ATP. ATP occupancy, PIP2 occupancy experiments are well described.

K39R leads to transient neonatal diabetes. In this paper, MD simulation proves that K39R increases the strength of the interaction of residue to PIP2. These data explain clearly the mechanism by which the mutation impairs insulin secretion and leads to neonatal diabetes. More over electrophysiology experiments shows that the introduction of K39R mutation into Kir6.2 increases the IC50 for nucleotide inhibition by about 1.5-3 fold.

PCF was then performed on ANAP-labeled Kir6.2* construct (on W311) in order to estimate the binding affinity of TNP-ATP for each of the Kir6.2*-K39 mutants (A,E and R) using FRET experiments coupled to electrophysiology. However the results are difficult to interpret . Fits of the MWC-type model shows that it was not possible to distinguish any change in the open probability of the channel.

These data are very interesting, the data are strong.

We thank reviewers for positive comments.

I have minor comments

Figure 1 : K69 should be P69

We thank the reviewer for the comment. We have now fixed the typo in the figure.

Figure 5 : Concentration of ATP are given in M (Should be added)

We thank the reviewer for the comment. We have now added the edit the units in the figure 5A.

Figure 7 : WT-ATP shows that K39 interacts with alpha and beta P of the ATP. This is in contradiction with the text line 14 page 6 : "K39 interacts with gamma P " Should be corrected In the legend it is mentioned an arrow « The arrow represents the change in motion of K39". Where is the arrow on the figure?

We thank the reviewer for the comment. We have now fixed the detail in the figure.

The PCF experiments : Could you explain why the W331 was chosen for the position of the ANAP. A supplementary figure of the structure of Kir6.2 with the locations of W311 (ANAP) and the TNP-ATP should be provided along with the location of the R39 mutants. This will be useful to understand the FRET experiments.

We thank the reviewer for the suggestion. We choose this position for the ANAP label based on our previous finding (Usher et al, 2020), foremost position 311 is 26 Å from the location of the inhibitory nucleotide-binding site (PDB accession #6BAA). This is so less likely to affect binding directly. We have now added a brief sentence describing this logic in the results section.

Referee #4:

NB Authors will see that the review below was written for previous submission of this manuscript to a different journal, but comparison of the two manuscripts indicates they are identical, so comments below should all be addressed in any revision.

We have substantially revised the manuscript from the one submitted to Biorxiv. Here, we will highlight our edits and corrections based on previous comments.

General:

The authors have used computational and experimental analyses to assess the interplay between ATP inhibition and PIP2 activation of KATP channels. This is a key molecular area of regulation of these channels and one that has been extensively studied previously. The authors bring some novel approaches to the issue, but there are concerns with the approach and interpretation, as detailed in the comments below.

Major:

1. A major concern regarding the modeling is that the results are described as facts, rather than testable predictions. This is a fundamental issue that needs to be acknowledged.

We thank the reviewer for the suggestion. We have now rephrased the manuscript throughout on those ideas.

2. Experimentally, there is major concern that, as the paper mentioned, there are

inconsistencies between ATP inhibition and TNP-ATP inhibition. The main potential novelty of the modeling lies in identifying K39 as a residue that can interact with phosphate groups of both ATP and PIP2. The authors acknowledge (Fig. S7) that the TNP group may interfere with the binding at K39, but the FRET assay and model are both for TNP-ATP which may not be able to explain the mechanism of ATP inhibition of the current.

We thank the reviewer for reading the previous submission of the manuscript. Note that we do not have Fig S7 which is described in the comment in this current revision of the manuscript submitted to JPhysiol.

3. The authors used MD simulations and experimental fluoro-patching to see how certain mutations affect ATP binding and channel activity, from which molecular mechanisms of apparent exclusive binding of PIP2 and ATP to KATP proteins may be inferred. Based on their MD simulations, the authors claim K39 is the key residue to facilitate PIP2 binding over ATP and that increased hydrogen bonds to PIP2 explains gain of function of the K39R mutant. Experimentally they tried to correlate the changes in ATP binding and ATP inhibition of several mutant channels, although this was not very successful. First, the IC50 for TNP-ATP current inhibition and EC50 of TNP-ATP FRET differed by 2 orders of magnitude, making it impossible to directly correlate TNP-ATP binding to functional modulation. Second, the functional results regarding K39 mutants are not in accordance with their simulation results: simulations imply that hydrogen bonding between the K39 amine and PIP2 is critical for PIP2 binding. If this is correct, then K39A and K39E should give a loss of function phenotype, which is not observed. K39E shows a gain of function phenotype, which is even stronger in the absence of SUR1, best mimicking the MD simulations. This questions whether K39R gain of function effect can be attributed to increased interaction with PIP2? Could it be a gating mutant with increased open state stability independent of PIP2 binding?

We thank the reviewer for reading the previous submission of the manuscript. However, this sentence has been thoroughly revised in the current submission to JPhysiol.

4. There seems to be confusion regarding previous studies that have analyzed the interaction of ATP and PIP2 in regulating the channel (p.3 last paragraph). The authors say "Because an increased channel open probability is associated with reduced ATP inhibition^{20,21}, it is possible that at least part of the effect of PIP2 is mediated via changes in Popen. However, it has also been argued that PIP2 may have an additional effect on ATP sensitivity that is independent of Popen²⁰". On p6 they say "Previous studies have shown that PIP2 reduces channel ATP inhibition^{4,19,30,31}. However, it was not clear if PIP2 competes directly for the ATP binding site or if it interferes with ATP dependent gating (or both)." These statements misrepresent previous studies, particularly multiple studies by Enkvetchakul/Nichols/Shyng groups [including ref 20, 30, and unreferenced Biophys J. 2001;80(2):719-28; J Gen Physiol. 2003 Nov;122(5):471-80]. Those studies mechanistically explain how the change in Popen resulting from PIP2 binding causes loss of ATP sensitivity, and do not suggest any additional effect on ATP sensitivity: the first paper to report PIP2 modulation of both Popen and ATP was ref 30, which specifically discussed how 'negative heterotropic cooperativity' between the two ligands (meaning that they both compete for the same unliganded channel, without having to bind at the same site) could explain the effect the effect of PIP2 on ATP sensitivity as a direct consequence of the effect on Popen, as was subsequently quantitatively confirmed and modeled by Enkvetchakul et al.

We thank the reviewer for reading the previous submission of the manuscript. However, this sentence has been thoroughly revised in the current submission to JPhysiol.

5. The authors go on to say "previous studies have proposed that PIP₂ competes with ATP for the same binding site on the C-terminus of the protein²². However, comparison of recent structural studies of the channel with bound ATP^{5,6}, and docking and molecular dynamics simulations with PIP₂ suggest that ATP..." Although the authors quote ref 22 as having proposed that PIP₂ and ATP compete for the same binding site, that study was carried out on Kir1.1, which is not a KATP channel. Since the Enkvetchakul studies show how 'competition' between PIP₂ and ATP arises without the two ligands binding in the same pocket, the point being made about the binding pockets being different is not an argument against competition. The authors describe what they consider to be two alternate concepts for how PIP₂ affects ATP sensitivity, but they are really the same - PIP₂ competes with ATP for binding to the unliganded channel, what the authors describe as a "local allosteric effect" is the same as 'negative heterotropic cooperativity'. Even though PIP₂ and ATP may not bind at the same site (the sites could be far apart), ATP binding will still be reduced if PIP₂ is increased, because the fraction of unliganded channels will be reduced.

We thank the reviewer for reading the previous submission of the manuscript. However, this sentence has been thoroughly revised in the current submission to JPhysiol.

6. How valid or reliable is the PIP₂ bound structure, derived from CG-MD? How can this be reconciled with the fact that Lys170 at the bottom of the TM2 and E179, both critical for PIP₂ gating, are not directly interacting with PIP₂ at all in the simulations? Also, the mode of PIP₂ binding to Kir channels is quite different from what has been observed in many crystal structures where the 5' P makes more extensive interactions with the neighboring basic residues while the 4' P makes limited interactions and faces away from the protein?

We thank reviewer for the comments. We converted PIP₂ from di-C8 PIP₂ in the chicken structure to the CG representation to equilibrate the binding pose to fit with the Kir6.2, then re-convert the structure back to atomistic, and then further equilibrate the binding pose of PIP₂ within the binding site for an additional 80 ns. We conduct those multi-step conversions to simply adjust the binding pose of PIP₂ to fit with the Kir2.2. To confirm our binding site, we have aligned the position of the inositol group from diC8:PIP₂ chicken Kir2.2 structure and relax the position of PIP₂ using similar equilibration protocols. We compare the position of PIP₂ during the final 15 ns of our equilibration where the position of PIP₂ is unrestrained, with the final snapshot from our 380ns simulation (Supplementary figure 3). This alignment shows that our binding site is within the conformational space of PIP₂ obtained from the crystal structure. These information are now included in supplementary figure 3. Both K170 and E179 are not in contact with PIP₂ in our simulations. It also did not make any contact in an independent simulation conducted by Stary-Weinzinger group, which sample this interaction for 1 μ s using Amber forcefield.

7. Why was E179 not analyzed in MD trajectories in the same way as done for K39? The gain of function effect of E179A or E179K is even greater than K39R, and the kinetic model suggests that E179A and E179K also reduce the nucleotide binding affinity.

We thank the reviewer for reading the previous submission of the manuscript. However, the aforementioned data mentioned in this comment is not in the current submission to JPhysiol.

Other comments

4. Fig1A residue labeling is incorrect: R54 and R176 should be switched, and the text describing that R54 and K67 from one subunit and R176 from an adjacent subunit is

incorrect.

We thank the reviewer for reading the previous submission of the manuscript. However, this figure has been corrected in the current submission to JPhysiol.

5. K67 and R176 are from the same subunit and R54 is from neighboring subunit.

We thank the reviewer for reading the previous submission of the manuscript. However, this figure has been corrected in the current submission to JPhysiol.

What is the basis for considering RMSF > 1Angstrom to be biologically significant?

We thank the reviewer for reading the previous submission of the manuscript. However, this clarity has now been corrected in the current submission to JPhysiol.

6. The amine group of Lys residues is better described as a terminal amine group rather than a head group. A head group is the term used to describe lipid structures, and it is unconventional to call an aa side chain part a head group.

We thank the reviewer for reading the previous submission of the manuscript. We have now addressed the notation of the term headgroup in this revision.

7. The finding that K39 may interact with either ATP or PIP2 is very interesting and suggests it may actually be involved in both binding sites. However, cryo-EM structures have only shown CTD-disengaged conformations for Kir6.2 (as opposed to the 'engaged' conformations that are also seen in Kir2.2 and Kir3.2 structures and which likely represent the active conformations), which results in a quite a distance between the PIP2 and ATP binding site of about 25 Å as mentioned in the manuscript. Presumably the simulations involve 'engagement' and the binding sites are not that far apart in these simulations? Showing the distance between the two sites in their simulations will help the reader to understand how one of residue can interact with both substrates.

We thank reviewer for the concern. This is a very interesting point to make. To address the point, we simulated the open state structure where ATP is not bound, and the structure is opened. We observed no difference in PIP₂ contacting residues.

We suspect that the Kir6.2 is always in the "engaged" state, whether it is closed or open. This is quite distinct when you compare Kir6.2 to Kir6.1, the position CTD is much more engaged in Kir6.2. This idea has been proposed by MacKinnon that PIP₂ binding in Kir6.2 is not necessary for the transition to the engaged state, and thus, this engagement event is plausible and is only plausible in Kir6.2.

8. Fig 2A R54 is mislabeled and it is likely to be R176.

We thank the reviewer for reading the previous submission of the manuscript. However, this clarity has now been corrected in the current submission to JPhysiol.

9. Would the increased H-bonds between K39R and PIP2 also increase H-bonds between K39R and ATP? If the mutation increased interaction for both substrates, this would argue against the gain of function phenotype of the K39R mutant being due to increased PIP2 affinity. Therefore, it is necessary to show how the H-bonding pattern differs for K39 versus K39R and ATP. The postulate, 'K39R will lead to reduced channel inhibition by ATP, and thereby impairs insulin secretion leading to neonatal diabetes' is questionable.

We thank reviewer for the concern. We have now analysed H-bond frequency between K39 and ATP as well as PIP₂. These information are now shown in Fig S8B.

10. In the discussion; the authors suggest that steric clashes may interfere with TNP-ATP binding to K39R mutant. It is difficult to imagine that the K to R mutation could create much difference and, in addition, K39R shows yet stronger binding (IC₅₀ = 2.62 μ M) to the site than K39A (IC₅₀= 13.2 μ M). If steric occlusion was the cause, this should be relieved with the smaller sidechain in the K39A mutation and the binding should then be stronger?

We thank reviewer for a very interesting question in the discussion. We have revised the manuscript discussion in the previous revision and hence, this point of discussion has now been removed from the latest version of the manuscript submitted to JPhysiol.

11. Ref 14 and 27 are the same.

We thank the reviewer for reading the previous submission of the manuscript. However, this clarity has now been corrected in the current submission to JPhysiol.

Dear Dr Stansfeld,

Re: JP-RP-2022-283345X "The dynamic interplay of PIP2 and ATP in the regulation of the KATP channel" by Tanadet Pipatpolkai, Samuel G Usher, Natascia Vedovato, Frances M. Ashcroft, and Phillip Stansfeld

I am pleased to tell you that your paper has been accepted for publication in The Journal of Physiology.

NEW POLICY: In order to improve the transparency of its peer review process The Journal of Physiology publishes online as supporting information the peer review history of all articles accepted for publication. Readers will have access to decision letters, including all Editors' comments and referee reports, for each version of the manuscript and any author responses to peer review comments. Referees can decide whether or not they wish to be named on the peer review history document.

The last Word version of the paper submitted will be used by the Production Editors to prepare your proof. When this is ready you will receive an email containing a link to Wiley's Online Proofing System. The proof should be checked and corrected as quickly as possible.

Authors should note that it is too late at this point to offer corrections prior to proofing. The accepted version will be published online, ahead of the copy edited and typeset version being made available. Major corrections at proof stage, such as changes to figures, will be referred to the Reviewing Editor for approval before they can be incorporated. Only minor changes, such as to style and consistency, should be made a proof stage. Changes that need to be made after proof stage will usually require a formal correction notice.

All queries at proof stage should be sent to TJP@wiley.com

Are you on Twitter? Once your paper is online, why not share your achievement with your followers. Please tag The Journal (@jphysiol) in any tweets and we will share your accepted paper with our 23,000+ followers!

Yours sincerely,

Ian D. Forsythe
Deputy Editor-in-Chief
The Journal of Physiology
<https://jp.msubmit.net>
<http://jp.physoc.org>
The Physiological Society
Hodgkin Huxley House
30 Farringdon Lane
London, EC1R 3AW
UK
<http://www.physoc.org>
<http://journals.physoc.org>

P.S. - You can help your research get the attention it deserves! Check out Wiley's free Promotion Guide for best-practice recommendations for promoting your work at www.wileyauthors.com/eeo/guide. And learn more about Wiley Editing Services which offers professional video, design, and writing services to create shareable video abstracts, infographics, conference posters, lay summaries, and research news stories for your research at www.wileyauthors.com/eeo/promotion.

*** IMPORTANT NOTICE ABOUT OPEN ACCESS ***

To assist authors whose funding agencies mandate public access to published research findings sooner than 12 months after publication The Journal of Physiology allows authors to pay an open access (OA) fee to have their papers made freely available immediately on publication.

You will receive an email from Wiley with details on how to register or log-in to Wiley Authors Services where you will be able to place an OnlineOpen order.

You can check if your funder or institution has a Wiley Open Access Account here <https://authorservices.wiley.com/author-resources/Journal-Authors/licensing-and-open-access/open-access/author-compliance-tool.html>

Your article will be made Open Access upon publication, or as soon as payment is received.

If you wish to put your paper on an OA website such as PMC or UKPMC or your institutional repository within 12 months of

publication you must pay the open access fee, which covers the cost of publication.

OnlineOpen articles are deposited in PubMed Central (PMC) and PMC mirror sites. Authors of OnlineOpen articles are permitted to post the final, published PDF of their article on a website, institutional repository, or other free public server, immediately on publication.

Note to NIH-funded authors: The Journal of Physiology is published on PMC 12 months after publication, NIH-funded authors DO NOT NEED to pay to publish and DO NOT NEED to post their accepted papers on PMC.

EDITOR COMMENTS

Reviewing Editor:

I am satisfied with the revisions.

REFEREE COMMENTS

Referee #3:

The authors have responded satisfactorily to all the points I have raised

Referee #4:

None further.